# Exploratory Metabolomics and Lipidomics Profiling Contributes to Understanding How Curcumin Improves Quality of Goat Semen Stored at 16 °C in Tropical Areas

**DOI:** 10.3390/ijms251810200

**Published:** 2024-09-23

**Authors:** Zhaoxiang An, Liguang Shi, Hanlin Zhou, Guanyu Hou, Wenjuan Xun

**Affiliations:** 1Tropical Crops Genetic Resources Institute, Chinese Academy of Tropical Agricultural Sciences, Haikou 571100, China; anzx1111@163.com (Z.A.); shiliguang123@126.com (L.S.); guanyuhou@126.com (G.H.); 2School of Tropical Agriculture and Forestry, Hainan University, Haikou 570228, China; 3Zhanjiang Experimental Station, Chinese Academy of Tropical Agricultural Sciences, Zhanjiang 524000, China; zhouhanlin8@163.com

**Keywords:** curcumin, goat semen storage, oxidative stress, metabolomics and lipidomics

## Abstract

Reactive oxygen species (ROS) exert a vital role in sperm quality during semen preservation, where excessive ROS leads to oxidative damage and undermines sperm integrity. Curcumin, a botanical extract, is capable of neutralizing ROS and enhancing the activity of antioxidant enzymes. This study was aimed at evaluating the effects of curcumin on sperm viability, acrosome integrity, and antioxidant levels, as well as metabolomic and lipidomic profiles. The results demonstrated that curcumin at 25 µmol/L significantly enhanced sperm motility, plasma membrane, and acrosome integrity, elevated the levels of antioxidant enzymes (T-AOC, CAT, SOD), and decreased ROS production (*p* < 0.05). Metabolomic analysis identified 93 distinct metabolites that showed significant differences between the control and curcumin-treated groups. KEGG pathways emphasized the participation of these metabolites in key metabolic processes such as the citric acid cycle, cholesterol metabolism, and fatty acid metabolism. Curcumin treatment brought about notable variations in lipid profiles, including increased levels of phosphatidylcholine, acylcarnitine, and triglyceride over the storage time, suggesting enhanced lipid anabolic activity. Overall, the supplementation of curcumin at 25 µmol/L effectively mitigates oxidative stress and prolongs the viability of semen storage at 16 °C by modulating specific metabolic and lipid profiles.

## 1. Introduction

The success of artificial insemination in goats is determined by the quality of semen. The purpose of preserving semen is to reduce sperm metabolic activity while extending their survival in specialized environments. This preservation process aims to improve the efficiency and effectiveness of artificial insemination techniques in goat breeding [1,2]. The storage of semen at room temperature without requiring complex equipment is a simple operation that can only fulfill the short-term storage needs of various livestock. By adding acidic substances to the dilution agent in order to lower the pH, the metabolic activity of sperm is inhibited, thus maintaining the sperm in a reversible resting state and ultimately prolonging their survival in vitro [2]. Although the research on dilutive cryopreservation and the freezing process has been very in-depth [3], the resulting damage to ultrastructure and thermal stability remains a major concern [4,5]. Preservation at room temperature can effectively reduce damage, but as preservation time extends, sperm’s own production of reactive oxygen species (ROS) exceeds its antioxidant level, leading to oxidative stress and apoptosis that ultimately affects the preservation effectiveness [6]. The Hainan black goat is one of the meat breeds raised in hot areas of south China, where small-scale grazing is the main breeding mode. During the breeding season, the preservation and use of frozen semen is not commonly used due to the limitations of environmental high temperature and technical amenities, so it is more common to use easy-to-operate liquid dilution and storage at room temperature. Therefore, it is crucial to identify a method that can effectively preserve semen at room temperature for an extended period while maintaining high quality.

Lipid peroxidation (LPO) is the process of oxidizing polyunsaturated fatty acids (PUFA) in biological systems, leading to alterations in the structure and properties of the cell membranes. Sperm are particularly susceptible to peroxidative damage due to their high content of polyunsaturated fatty acids, which significantly impacts their fertilization ability [7,8]. As a result, studies have demonstrated that adding exogenous antioxidants to the diluent can effectively reduce oxidative damage and improve semen preservation outcomes [9,10].

Curcumin (Cur) is a natural product predominantly found in turmeric, possessing diverse biological activities such as antioxidant, anti-inflammatory, anti-tumor, and antibacterial effects [11,12]. Cur has the capability to eliminate reactive oxygen species (ROS), including superoxide and hydroxyl radicals [13]. Additionally, it can also inhibit LPO [14]. While curcumin has been shown to effectively enhance semen quality after cryopreservation [15,16,17], limited research exists on its protective effect on semen at 16 °C.

At the same time, omics technology has been widely utilized in the field of biology due to its continuous development. The maturation of methods such as genomics, metabolomics, transcriptomics, and lipidomics has expanded possibilities for understanding complex processes within living organisms [18]. In studying semen quality and preservation ability, the application of omics technology can offer a more comprehensive and in-depth analysis covering biomolecular, cellular, and genetic information. This approach facilitates thorough exploration of differences in sperm preservation and associated influencing factors or biomarkers. Metabolomics, as the closest omics type to the organism phenotype, provides a comprehensive method for studying the metabolic characteristics of biological samples by analyzing the composition and changes of metabolites such as lipids, amino acids, and sugars in semen [19]. Discovering metabolic indicators related to preservation capacity allows for better understanding of potential biomarkers at 16 °C and reveals their relevant metabolic pathways and regulatory mechanisms. To fully comprehend the key factors affecting reproductive ability, lipidomic methods have become one of the most advanced research approaches. Lipidomics is dedicated to studying lipid profiles (lipidome) in cells, tissues, or organisms through qualitative and quantitative analysis of their lipid content. Sperm cells contain various lipid components that each have a unique effect on overall cell function. Lipidomics has been successfully applied to explore aspects regarding fatty acid content in sperm cells from different species [20]. Individual differences often lead to inconsistencies in preservation techniques at normal temperatures for goat semen; addressing these limitations is necessary for further improvement. In this study, the combined analysis of metabolomics and lipidomics revealed the differential performance of semen samples of different quality after storage at 16 °C. This provides a research basis and possible development direction for further improving the normal temperature preservation technology of Hainan black goat semen.

The aim of this study was to investigate whether curcumin can enhance sperm antioxidant capacity and reduce levels of reactive oxygen species and malondialdehyde content in Hainan black goats. In addition, significant differences in metabolomic and lipidomic characteristics were compared among semen samples from different groups to elucidate the metabolic pathways and key markers that may contribute to these differences. The results of this study will facilitate a deeper understanding of the molecular mechanisms underlying black goat sperm fluid preservation capacity. Furthermore, it will provide relevant information and guidance for optimizing semen preservation techniques and quality assessment methods.

## 2. Results

### 2.1. The Impact of Curcumin Supplementation on Sperm Kinetic Parameters

The effects of different concentrations of curcumin on the kinetic parameters of Hainan black goat sperm during liquid storage at 16 °C are presented in Table 1. Compared to the control group, the addition of Cur significantly increased VSL at 48 h only (*p* < 0.05), while it decreased VSL at 72 h in the 15 and 50 μmol/L groups. The VCL, VAP, and VSL in the 25 μmol/L group were the highest among all treatment groups, but there was no significant difference between the 25 mm group and control group (*p* > 0.05).

### 2.2. The Impact of Curcumin on Sperm Quality When Preserved at 16 °C

Figure 1A illustrates the impact of different concentrations of curcumin on sperm motility. The group with a concentration of 25 μmol/L consistently showed a strong protective effect in each period, and this difference was statistically significant compared to the control group (*p* < 0.05). In addition, the other treatment groups also experienced an improvement in sperm motility. After a 24 h period of storage, no notable variations were observed in the plasma membrane integrity rate among the remaining groups, except for the group exposed to 25 µmol/L, which showed a statistically significant difference (*p* < 0.05). After being stored for 72 h, there was no noticeable distinction between the group with a concentration of 50 μmol/L and the control group. However, the group with a concentration of 25 μmol/L showed the most favorable therapeutic outcome. During the first 24 h, there were no noticeable variations in acrosome integrity across all groups. However, between 48 and 72 h, a notable distinction was found between the 5 and 25 μmol/L groups and the control group, with statistical significance (*p* < 0.05).

### 2.3. Effect of Curcumin on the Antioxidant Properties of Sperm Preserved at 16 °C

Figure 2 illustrates the impact of varying doses of Cur on the antioxidant characteristics of Hainan black goat semen during 16 °C liquid storage. The effects on CAT activity are depicted in Figure 2A. At 72 h, the CAT activity in the groups treated with 5, 15, and 50 μmol/L of semen was not substantially different from the control group (*p* > 0.05). However, the CAT activity was highest in the group treated with 25 μmol/L of semen, and this difference was statistically significant compared to the other groups (*p* < 0.05). The T-AOC activity of the semen exhibited an initial upward trend followed by a subsequent decline. At the 72 h mark, there was a significant difference between the control group and the 25 μmol/L group (*p* < 0.05). However, there was no significant difference between the groups with concentrations of 5, 15, and 50 μmol/L (*p* > 0.05) (Figure 2B). Following 72 h of preservation, the group treated with 25 μmol/L showed a considerably higher level of GSH-Px activity compared to the other groups (*p* < 0.05). However, there was no significant difference in GSH-Px activity between the groups treated with 15 and 50 μmol/L (*p* > 0.05) (Figure 2C). In Figure 2D, there was no significant change in MDA content between the 15 and 50 μmol/L groups (*p* > 0.05). However, the 25 μmol/L group had considerably reduced MDA content compared to the other groups (*p* < 0.05). Following 72 h of preservation, the group treated with 25 μmol/L exhibited a considerably increased level of SOD activity compared to the other groups (*p* < 0.05) (Figure 2E). After 72 h, the levels of ROS at concentrations of 5, 15, 25, and 50 μmol/L were significantly lower compared to the control group (*p* < 0.05) (Figure 2F).

### 2.4. Effect of Curcumin on Mitochondrial Function When Preserved at 16 °C

Figure 3A illustrates the impact of varying concentrations of Cur on ATP levels during sperm preservation at a temperature of 16 °C. At 24 and 48 h, the levels of ATP in sperm were substantially greater in the groups exposed to 25 and 50 μmol/L compared to the control group and the groups exposed to 5 and 15 μmol/L (*p* < 0.05). The maximum MMP was observed in the group with a concentration of 25 μmol/L. However, after 48 h, there was no significant difference between the 25 μmol/L group and the 50 μmol/L group (*p* > 0.05).

Figure 3B shows that the MMP was highest in the group treated with 25 μmol/L, and this value differed substantially from the control group and the groups treated with 5 and 50 μmol/L (*p* < 0.05). However, there was no significant difference between the group treated with 25 μmol/L and the group treated with 15 μmol/L (*p* > 0.05).

### 2.5. Metabolite Profiles in Sperm

In order to better understand the patterns of metabolite changes in various curcumin treatments, the primary and secondary metabolites in the samples were identified using UP target-targeted metabolome technology on the UPLC-MS platform. Figure 4A,C clearly shows that the data from different treatments are spread out on both sides of the confidence interval. This indicates a distinct separation along the principal component axis. In addition, BD presents the cross-validation results of the model in Appendix A. The R2Y (cum) parameters are 0.995 and 0.995, while the Q2 (cum) values are 0.819 and 0.426. The data detection confirms that the model demonstrates high interpretability and predictability.

### 2.6. Cluster Analysis of the Differentially Expressed Metabolites

In order to categorize all the discovered metabolites in sperm, the metabolites were matched to the database. The research identified 820 characteristics that were annotated using the HMDB database. Lipids and lipid compounds were the main metabolites, accounting for 24.9% of all detected metabolites. Additional noteworthy categories of metabolites comprised of organic acids and derivatives (22.3%), organoheterocyclic compounds (14.9%), organic oxygen compounds (10.5%), and benzenoids (10.1%). In addition, phenylpropanoids, polyketides, and eight additional metabolites accounted for the remaining 17.3% (Appendix A).

Analyzing the clustering heatmap (Figure 5) allows for the examination of variations in the accumulation patterns of metabolites across different samples. The cluster heatmap analysis revealed obvious differences in substances among different groups and also demonstrated equal clustering between different biological replicates, demonstrating strong homogeneity among biological replicates and great data dependability. In the control group, there was an increase in the levels of malonic acid, sucrose, 2-deoxyribose-5-phosphate, ethanolamine, palmitoylcarnitine (Car (16:0)), Car (18:0), and inosine. The following amino acids were shown to be elevated in the diluent group: proline, lysine, norleucine, alanine, beta-alanine, malic acid, and isoleucine. The compounds in the curcumin group are taurocholic acid, tyrosine, glyoxal, isocitric acid, succinate, and leucine.

### 2.7. Screening and Identification of Differential Metabolites

After analyzing the differential metabolites in the diluent group, it was discovered that all of these metabolites showed an increase in levels of organic acids and derivatives, alkaloids and derivatives, polyketides, and amino acids and peptides. In contrast, lipids and lipid-like compounds as well as nucleosides, nucleotides, and analogues were found to be downregulated, as shown in Figure 6A. In the Cur group, the expression of alkaloids and their derivatives was reduced, as seen in Figure 6B. Figure 6C,D display the 20 most significant metabolites that differ when comparing the various combinations of VIP levels. Notably, the diluent group exhibited modifications in amino acids and lipids, while the Cur group showed alterations in alcohol and amino acids among the significantly modified metabolites. We identified the top 10 differential metabolites that showed an increase or decrease in each comparison group based on fold change (FC) during metabolite accumulation (Figure 6E,F).

The screening parameters used in this work to discover differential metabolites were a *p*-value less than 0.05, a fold change larger than 1.5 or less than 0.67, and a VIP value exceeding 1. The volcano plot analysis identified a total of 250 metabolites that showed differential expression in the diluent group, with 125 being upregulated and 125 being downregulated (Figure 6G). In the Cur group, there were 93 divergent metabolites, with 46 being upregulated and 47 being downregulated (Figure 6H). Several substances, including malate, subtaurine, succinate, pantothenic acid, isocitrate, citric acid, β-alanine, xanthosine5-phosphate, nicotinamidemononucleotide (NMN), xanthine, nobiletin, nicotinamide riboside (NR), hypotaurine, tyrosine, taurocholic acid, oleuropein, inosine, valproic acid, and others, were found to coexist in certain pathways, indicating their potential importance during the preservation process. For more details, please refer to Appendix A.

### 2.8. KEGG Enrichment Analysis of Differential Metabolites

Figure 7 demonstrates the analysis of differential metabolites to identify the pathways they are linked to. The impact of the route effects on general metabolism was assessed using pathway topology analysis, as shown in Appendix A. The diluent group may impact various pathways, such as the citrate cycle (TCA cycle), cholesterol metabolism, beta-alanine metabolism, ABC transporters, pantothenate and CoA biosynthesis, and glyoxylate and dicarboxylate metabolism (Figure 7A,B). The metabolism of nucleotides, nicotinate and nicotinamide, purines, glyoxylate, and dicarboxylate was found in the Cur group’s metabolism. Additionally, the citrate acid cycle (TCA cycle) showed considerable enrichment, as shown in Figure 7C,D.

### 2.9. Identification of Lipid Species and Quantification and Multivariate Statistical Analysis

The performance characteristics of the OPLS-DA model found that the diluent group and Cur group all showed separation, and the permutation test (200 random permutations) verified the model; no overfitting of the data was observed (Figure 8A,B) (Appendix A).

After analyzing positive and negative ions using LipidSearch, 49 lipid molecules and 2945 lipid molecules were identified. The statistical results of the lipid subclasses identified by positive and negative ion patterns and the number of lipid molecules identified in various classes are shown in Appendix A. Phosphatidylserine, phosphatidylethanolamine, triglyceride, phosphatidylethanolamine, and triglyceride were the lipids with high content in each group (Figure 8C), and the relative abundance of each lipid in each group is shown in Figure 8D,E.

### 2.10. Screening and Clustering Analysis of Potential Lipid Molecular Markers

Initially, we conducted a comparative analysis of the lipid variations across the groups. Our findings indicate that there were substantial differences in LPC, LPE, and DG levels in the diluent group (Figure 9A), whereas AcCa, PE, and PS levels in the Cur group exhibited significant changes (Figure 9B). The study employed univariate statistical analysis techniques, including fold change analysis and t-test, to investigate lipid variations between groups and discover potential lipid molecular markers. The selection of potential markers was based on VIP scores obtained from OPLS-DA, as well as lipids with a fold change (FC) larger than 1.5 or less than 0.67 and a *p*-value less than 0.05. Volcano plots were generated, as depicted in Figure 9C,D. To evaluate the reasoning behind the differences in lipid molecules and visually display the connection between samples, as well as the changes in lipid expression patterns, the samples were categorized into clusters based on the expression of lipid molecules that exhibited notable distinctions (VIP > 1, *p* < 0.05). Table 2 displays the examination of a total of 46 lipids in the diluent group. Table 3 displays a selection of 30 lipids from the Cur group. Simultaneously, the association analysis of lipid molecules with notable disparities revealed that the lipid expression profiles could be clearly classified into two distinct categories. The Cur group demonstrated an increase in the expression of several lipid species, such as TG, PE, and DG, whereas phosphatidylinositols (PI) expression only increased in the control group (Figure 9C,D). The results indicate that there were significant differences in the lipid expression patterns of sperm between the diluent and Cur groups following preservation.

### 2.11. Correlation among Diverse Lipids

The objective of doing a differential lipid association analysis is to investigate the uniformity of lipid change patterns and evaluate the link between differentially expressed lipids using the Pearson correlation coefficient. The lipid correlations frequently uncover simultaneous modifications in lipids: when changes coincide with those of particular lipids, it signifies a positive correlation; conversely, if changes oppose those of specific lipids, it implies a negative correlation (Appendix A).

A strong positive connection (R = 0.852, *p* < 0.01) was seen between Hex2Cer (t33:2) and DG (18:0_22:5) in the diluent group. Furthermore, the compound PE (12:0e_6:0) displayed a remarkably strong positive correlation with LPC (15:0) (R = 0.999, *p* < 0.01). Similarly, TG (12:0e_8:0_14:0) revealed a highly significant positive correlation with both DG (16:0_20:3) and DG (18:0_16:00) (R = 0.999, *p* < 0.01). Moreover, there was a strong and statistically significant positive association (R = 0.999, *p* < 0.01) observed between PG (20:0_14:0) and PE (16:0e_20:2). Conversely, TG (16:0_16:1_18:1) exhibited a highly significant negative connection with TG (4:0_11:3_18:3) (R = −0.766, *p* < 0.01). Additionally, PC (29:1) demonstrated a considerably negative association with Hex2Cer (d16:0_20:3) (R = −0.746, p = 0.013) (Figure 10A,B).

In the Cur group, there were strong positive correlations between DG (24:5e) and AcCa (20:3) (correlation coefficient R = 0.999, *p* < 0.01), PE (12:0e_6:0) and LPC (correlation coefficient R = 0.999, *p* < 0.01), PE (12:0e_6:0) and LPE (correlation coefficient R = 0.996, *p* < 0.01), and PC (31:1) and PE (16:0_18:1) (correlation coefficient R = 0.996, *p* < 0.01). There was a significant negative correlation between PC (23:1_11:2) and PS (20:2e_20:4) (R = −0.734, *p* = 0.015). Similarly, there was a significant negative correlation between DG (18:3e_10:4) and PE (38:7e) (R = −0.727, *p* = 0.017). Additionally, there was a significant negative correlation between AcCa (22:4) and PC (23:1_11:1:2) (R = −0.721, *p* = 0.018). Lastly, there was a significant negative correlation between DG (18:3e_10:4) and PS (20:2e_20:4) (R = −0.705, *p* = 0.022) (Figure 10C,D).

## 3. Discussion

During preservation, the addition of exogenous antioxidants to extenders to resist oxidative stress has become the primary focus [21]. This approach effectively prolongs semen storage time while ensuring sperm quality [22,23]. The aim of this study was to investigate the protective effect of curcumin supplementation on goat semen and its potential mechanism. Specifically, the addition of 25 μmol/L curcumin increased sperm motility, preserved sperm plasma membrane and acrosome integrity, inhibited ROS production, and further extended sperm storage time at 16 °C by affecting the expression of metabolites and lipids.

The sperm plasma membrane, which serves as the outer membrane structure of sperm, plays a vital role in preserving the morphology, structure, and function of the sperm. It offers assistance and safeguarding while also being crucial for the movement of sperm. VSL, VCL, and VAP are important parameters for evaluating sperm kinematic parameters, and fertilization ability is closely related to these kinematic parameters. In this study, the addition of exogenous Cur significantly improved spermatozoa kinetic parameters compared to the control group. Curcumin also can significantly improve sperm plasma membrane integrity rate. The plasma membrane of sperm contains rich unsaturated fatty acids. When these react with superoxide anions, it leads to lipid peroxidation, resulting in loss of fatty acids in the plasma membrane and reductions in fluidity and integrity. Levels of LPO reaction products and MDA partly reflect oxidative damage in sperm [24,25]. Excessive MDA produced by oxidative damage inhibits SOD activity, eventually leading to decreased semen quality. This study discovered that curcumin effectively improved the integrity of the acrosome and reduced the level of MDA, resulting in enhanced SOD activity and ultimately enhancing the quality of semen.

ROS, including hydrogen peroxide (H_2_O_2_) and superoxide anion radical (O_2_^−^), are crucial redox signaling agents that are produced by over 40 enzymes under the regulation of growth factors and cytokines. These enzymes mainly include NADPH oxidase and the mitochondrial electron transport chain [26]. Several main pathways contribute to ROS production in semen. The metabolic activity of sperm is a major source of reactive oxygen species (ROS), which is generated by several routes including the mitochondrial respiratory chain, catalase, and phosphocreatinase. Furthermore, the engagement of sperm with other cells or molecules also results in the generation of intermediate levels of ROS. In addition, external factors such as environmental pollution can also lead to the generation of ROS [27,28,29]. Semen possesses some antioxidant capacity; among these capacities, the first-line defence mechanism called the enzyme triad is the most effective. This involves antioxidant enzymes such as superoxide dismutase (SOD), catalase (CAT), and glutathione peroxidase (GSH-Px) [30]. However, with prolonged preservation duration, there was a significant increase in ROS levels. The presence of CAT and GSH-Px plays a protective role against sperm damage caused by ROS. Curcumin has been found to decrease ROS production and enhance the antioxidant capacity of sperm [13,31]. In this study, the addition of curcumin led to a notable increase in total antioxidant capacity (T-AOC), reduced ROS production, and improved sperm quality. This finding is consistent with previous studies [32,33].

The mitochondrial membrane potential and ATP content play essential roles in sperm function. The mitochondrial membrane potential is a key factor in maintaining mitochondrial function and ATP synthesis [34]. It directly affects the proton gradient and the activity of ATP synthetase during oxidative phosphorylation [35]. High mitochondrial membrane potential is typically associated with greater sperm motility and viability, while low mitochondrial membrane potential may result in impaired energy metabolism and abnormal sperm function [36]. Since ATP serves as the primary cellular energy source, it is crucial for sustaining sperm motility, fertilization capacity, and other vital functions. After 72 h of preservation at a concentration of 25 μmol/L curcumin, high MMP was found to be beneficial for protecting mitochondrial function and ensuring ATP production, thereby collectively maintaining semen quality.

Several studies have explored the impacts of nano-formulations of diverse herbal extracts (MENFs, TENFs, and CENFs) on the cryopreservation quality of goat semen. The outcomes revealed that all nano-formulations were efficacious in enhancing sperm motility and membrane integrity, particularly CENFs at a concentration of 100 μg, which significantly ameliorated sperm parameters and antioxidant status [17]. This is analogous to discoveries in human semen, where 20 μM of curcumin notably enhanced sperm motility, decreased ROS and DNA fragmentation, and increased GPX4 gene expression [16]. Overall, the findings of this research are similar to those of the previous studies, indicating that the protective effect on sperm quality of supplementing with antioxidants like curcumin is continuously being investigated, which offers robust support for semen manipulation and preservation.

Cur supplementation markedly enhanced the survival rate of stored sperm, suppressed the generation of ROS, and conserved mitochondrial activity. Nevertheless, the precise mechanisms through which these effects are generated have not been completely clarified. The preservation of optimal conditions is intricately connected to the execution of metabolic functions. Metabolomics can be employed to evaluate certain metabolites, which can offer compelling evidence for explaining observed phenotypes. This work utilizes LC-MS non-targeted metabolomics to monitor changes in metabolites induced by curcumin during semen preservation. This study deepens our understanding of the possible protective mechanism of curcumin.

The study results reveal that differential metabolites were identified in various groups, and these metabolites were primarily associated with lipid metabolism, energy metabolism, and glucose metabolism. In particular, the tricarboxylic acid (TCA) cycle, as the fundamental energy metabolism pathway, plays a crucial role in sperm maturation and the maintenance of reproductive function [37,38]. Malic acid, as one of the key intermediates of the TCA cycle, can enhance citrate synthesis in the TCA cycle through a series of enzymatic reactions ultimately leading to ATP generation [39,40]. Additionally, malate may also safeguard sperm cell membrane and DNA from oxidative damage by inhibiting the oxidative stress response [41]. The study revealed a significant increase in malate content in the diluent group, suggesting an upregulation of malate levels. This could be attributed to the disruption of signaling pathways related to sperm energy supply, particularly those associated with the TCA cycle. Consequently, it is possible that sperm have reduced malate utilization and abnormal accumulation of malate in seminal plasma. Furthermore, the addition of curcumin may enhance sperm utilization of malate, thus promoting energy metabolism and ultimately ensuring the maintenance of semen quality.

Pantothenate participates in the synthesis of pantothenyl–coenzyme A, which promotes the normal progression of the energy metabolism pathway and provides essential energy for sperm to maintain motility [42]. Nicotinic acid has a protective effect by regulating cellular glycolysis to reduce sperm cell apoptosis [43]. Inositol plays a significant role in maintaining osmotic pressure, increasing sperm motility and oxygen consumption, as well as protecting sperm from DNA damage [44,45,46]. Leucine and isoleucine have been identified as potential biomarkers for semen preservation ability in boars [47], possibly affecting preservation through the biosynthesis of the amino acid pathway. Additionally, studies have reported positive effects of subtaurine on sperm motility in hamsters, cattle, chickens, and humans [48,49], associated with increased capacity for sperm liquid storage. Therefore, we speculate that malic acid, pantothenate, nicotinic acid, inositol, leucine, and isoleucine may serve as potential markers for reducing oxidative damage in sperm.

Prolonged preservation of semen leads to a variety of physiological changes and biochemical processes, including alterations in membrane components. In this study, we examined potential mechanisms for preservation and identified differential metabolites through metabolomics analysis. Specifically, we focused on the relationship between sperm lipids and sperm characteristics following preservation in Hainan goats. The sperm plasma membrane is rich in lipids, which play crucial roles in various functions [50,51]. Lipidomics not only allows for a comprehensive exploration of lipid molecule functions but also serves as an important tool for discovering biomarkers that indicate changes in lipid composition and expression [52,53]. Unlike polar metabolites such as amino acids and nucleotides, lipids exhibit diversity at three levels: class, subclass, and individual molecules. This diverse structure corresponds to different levels of lipid function. For instance, studies of lipid function typically emphasize changes at the subclass level because lipids generally operate within subclass groups. It can be challenging to distinguish the specific functions of individual lipid molecules within the same subclass. Conversely, studies focusing on markers tend to examine the expression levels of individual lipid molecules and their diagnostic capabilities.

The study results reveal that preserved sperm lipids consist of a substantial number of lipid subclasses, including phosphatidylcholine (PC), phospholipid lethanolamine (PE), ceramide (CER), sphingolipids (SM), fatty acids (FA), and acylcarnitine (AcCa). The composition of various lipids differs among distinct groups. The lipidomic analysis conducted in this study demonstrated variations in the patterns of lipid metabolism among different groups, with significant differences observed in the lipid molecules of PA, FA, TG, and AcCa. Furthermore, related studies support the notion that sperm cells exhibit active lipid metabolism, and that the concentration of free fatty acids in seminal plasma can serve as an indicator of sperm cell energy source [54].

The relative abundance of phosphatidylcholine (PC) was high in each group. Previous studies have suggested that the relative abundance of PC is associated with freeze–thaw sperm motility [55,56,57]. As a freezing-related marker, PC is believed to potentially impact sperm motility, membrane integrity, and lipid peroxidation processes [58]. The presence of PC in the sperm membrane further emphasizes its importance for maintaining sperm function [59], particularly in terms of membrane fusion, fluidity, and protein binding [60]. In the cryopreservation of goat semen, it has been demonstrated that the addition of substances rich in PC can effectively prevent cold shock and injury to the sperm, with egg yolk combined with PC yielding even more significant effects [61,62,63]. The varying relative abundance of PC among the groups in this study suggests that PC plays a role during preservation. This is likely because PC can integrate into the sperm membrane and effectively protect against structural damage during preservation while also being efficiently utilized in metabolic processes [64].

Acylcarnitines (AcCa), which consist of fatty acyl esters of carnitine, play crucial roles in the beta-oxidation of fatty acids in mammals, a process essential for energy generation during spermatogenesis. Carnitine coordinates a shuttle system [65], facilitating the transport of free fatty acids and acyl-CoA derivatives across mitochondrial membranes to produce adenosine triphosphate. At the same time, acyl groups combine with carnitine to form acylcarnitine (typically acetylcarnitine), providing readily available energy for spermatozoa during spermatogenesis. Previous research has shown significantly lower levels of acetylcarnitine in the semen of oligozoospermic infertile men compared to fertile controls [66]. Reduced urinary concentrations of acylcarnitines, including long-chain variants, may indicate diminished long-chain fatty acid beta-oxidation, leading to decreased energy levels in cases of oligozoospermic infertility. As a result, the subsequent energy demands necessary for sustaining normal sperm production may not be met [65]. Intriguingly, the curcumin-treated group demonstrated superior semen quality following preservation, yet presented lower levels of AcCa compared to the diluent group. We postulate that high-quality semen displays exceptional energy metabolism mechanisms and is more inclined towards the glycolytic pathway, thereby reducing reliance on AcCa. This efficient metabolic route is the key to maintaining sperm motility and motor performance. High-quality semen also exhibits outstanding antioxidant properties and can effectively handle intermediate metabolites, including the oxidation products of AcCa, reducing the potential harm of oxidative stress on sperm and thereby influencing AcCa concentration.

In our research, we observed that the Cur group sperm exhibited a higher concentration of triglycerides. However, it is important to note that triglycerides typically do not serve as constituents of plasma membrane lipids. In fact, elevated levels of triglycerides have been associated with detrimental effects on sperm [67]. For example, studies have highlighted that rats with diabetes display reduced sperm motility and antioxidant enzyme activity alongside heightened triglyceride levels in their sperm. Additionally, infertility in mice, particularly those with Slc22a14 (riboflavin transporter) knockout, is linked to triglyceride accumulation in sperm, coupled with notable deficiencies in fatty acid beta-oxidation. This condition is further evidenced by decreased levels of acylcarnitines from the TCA cycle [68]. Interestingly, our study found that the sperm of the Cur group exhibited superior quality during preservation. Thus, the rise in the relative abundance of triglyceride lipid molecules may be due to sperm absorption of lipids from thinners during preservation. Previous research has shown that exogenous lipids can adhere to the plasma membrane of sperm, creating a protective barrier against damage by altering the lipid composition on the cell membrane. These lipids are exchanged between the medium and the cell membrane, replenishing lost lipids and stabilizing against damage.

In addition, we also focused on the presence of PUFA in each group. Research has indicated that PUFA plays a crucial role in maintaining sperm structure and male fertility, as well as helping to preserve the integrity of the cell membrane lipid bilayer [69]. However, when the double bond in PUFA affects the methyl carbon–hydrogen bond, it can lead to oxidative damage and induce LPO, resulting in membrane abnormalities and decreased quality [70], even leading to loss of vitality [71,72]. Furthermore, overproduction of ROS during 16 °C preservation can cause altered carbohydrate, lipid, and protein composition in the membrane due to increased LPO. This oxidative damage, accompanied by loss of PC and PE [4], results in changes in the ultrastructure of sperm membranes. Cryopreservation has been shown to severely affect membrane structure [73], and this may also be true for 16 °C preservation.

## 4. Materials and Methods

The Hainan black goats used in this experiment were obtained from the Genetic Resource Preserving Farms of Hainan Province (Danzhou, China). Samples were collected in strict accordance with the Standards for the Protection and Utilization of Laboratory Animals of the People’s Republic of China. Animal handling protocols, experimental layouts, and approaches were approved by the Animal Care and Use Committee of Hainan University, Haikou, Hainan province, China (protocol code HNUAUCC-2022-000140).

### 4.1. Experimental Design

In this study, we aimed to study the effect of curcumin on the preservation of semen from Hainan black goats at 16 °C. The following studies were conducted:(1)The effect of curcumin on semen quality (sperm motility, sperm plasma membrane and acrosome integrity, and motor performance).(2)The effect of curcumin on the antioxidant capacity of semen.(3)Metabolomic sequencing of the effects of curcumin on semen (Con: fresh semen; 3day: semen preserved for three days after adding diluent only; 3day-cur: semen preserved for three days after adding 25 µmol/L Cur).(4)Lipidomic sequencing of the effects of curcumin on semen (Con: fresh semen; 3day: semen preserved for three days after adding diluent only; 3day-cur: semen preserved for three days after adding 25 µmol/L Cur).

### 4.2. Preparation of Diluent

Glucose, fructose, tris, streptomycin combination, dimethyl sulfoxide, and curcumin were acquired from Sorbolol (Beijing, China). The citrate–tris dilution (including tris 27 g/L, 13.75 g/L citrate, fructose 10 g/L, and 1 million IU streptomycin combination) was supplemented with curcumin solutions dissolved in dimethyl sulfoxide of varying proportions to reach the desired treatment group concentrations (5 μmol/L, 15 μmol/L, 25 μmol/L, and 50 μmol/L).

### 4.3. Collection and Dilution of Seminal Fluid

Fifteen Hainan black goats, aged between 1.5 and 2 years and in optimal physical condition, were chosen from a uniform rearing environment for the purpose of collecting semen using electrical stimulation. The freshly collected semen must satisfy the following conditions: absence of any unusual odor, a visually opaque white appearance, a minimum sperm motility of 80%, and a sperm concentration exceeding 1.5 billion per milliliter. For subsequent investigation, qualified semen samples were utilized after being mixed to eliminate any variance between individual goats. The semen was subsequently diluted at ratios ranging from 1:1 to 1:5, and then cooled and equilibrated in a refrigerator maintained at a temperature of 16 °C. Periodically, the semen was extracted and warmed up to evaluate its quality.

### 4.4. Detection of Sperm Quality

#### 4.4.1. Sperm Motility Performance

The collected semen was extracted and incubated at 37 °C for 2 min. Subsequently, 20 μL droplets were dispensed onto a preheated sperm counting plate, followed by the assessment of sperm motility and motion parameters (RCZ-200G sperm quality imaging system, Shi Mengde Medical Technology Co., Ltd., Jiangsu, China) (VSL—linear motion speed, VAP—average pathway velocity, and VCL—curve motion speed) using a computer-assisted sperm analyzer. Each group randomly selected three fields with no less than 200 sperm assessed in each field. All experiments were conducted with a minimum of three replicates.

#### 4.4.2. Integrity of the Plasma Membrane

The hypo-osmotic swelling test (HOST) was used to assess the integrity of the sperm plasma membrane. The hypo-osmotic solution is prepared by dissolving 0.49 g of sodium citrate and 0.9 g of fructose in 100 mL of distilled water. In preparation for the test, a 20 µL portion of conserved semen is combined with 200 µL of the hypo-osmotic solution in a test tube. The mixture is then incubated at a temperature of 37 °C for a duration of 30 min. After vigorously shaking, 5 µL of the suspension is placed onto a slide, and 200 cells, distinguished by their swollen and non-swollen tails, are observed under a 400× phase-contrast microscope. Cells with enlarged tails are categorized as having intact cell membranes, whereas cells with non-enlarged tails are deemed to have compromised cell membranes.

#### 4.4.3. Integrity of the Acrosome

Acrosome integrity was assessed using Coomassie brilliant blue staining. To prepare the Coomassie brilliant blue dye, the container was washed with 95% ethanol before adding 100 mg of Coomassie brilliant blue G-250, which was then dissolved in 50 mL of 95% absolute ethanol. The resulting solution should appear blue. Next, 100 mL of 85% phosphoric acid was added, giving a brown or brown-red color. A sample containing 50 µL of preserved semen and 1 mL of 4% paraformaldehyde were mixed in a test tube and allowed to fix at 16 °C for 10 min. After centrifugation at 1500× g for 5 min, the supernatant was discarded, and a smear was made using 10 µL of semen. Following air-drying, the smear was stained with Coomassie brilliant blue dye for 30 min, then rinsed with water and air-dried again. Finally, under a magnification of ×1000 using an oil immersion lens, sperm were counted as showing acrosome integrity when the heads were observed to be stained blue, while unstained heads showed non-integrity.

### 4.5. Detection of the Antioxidant Properties

The BCA protein assay kit (P0009, Nanjing Jiancheng Co., Ltd., Nanjing, China) was used to measure the concentration of superalbumin, and the resulting data were used to calculate the antioxidant index. Afterwards, the test kits (A015-3-1, A001-1-2, A005-1-1, A007-1-1; Nanjing Jiancheng Co., Ltd. China) were utilized in accordance with the given instructions. The levels of T-AOC, T-SOD, GSH-Px, CAT, ROS, and MDA were quantified at wavelengths of 593 nm, 550 nm, 412 nm, 405 nm, 525 nm, and 532 nm, respectively.

### 4.6. Mitochondrial Function Assays

The ATP content in the goat sperm samples was determined using an ATP assay kit (Nanjing Jiancheng Co., Ltd., Nanjing, China). Briefly, the sample was centrifuged to remove the supernatant, and then the necessary working solution was prepared according to the manufacturer’s instructions. The luminosity value of each working solution was measured at 636 nm, and the ATP content was calculated using a specified formula.

Mitochondrial membrane potential (ΔΨ m) detection was conducted using a kit from Nanjing Jiancheng Co., Ltd. (Nanjing, China). The samples were centrifuged as directed, and the sperm were stained with a mixed JC-1 buffer before being incubated for 30 min at 37 °C. After centrifugation at 800 g for five minutes to remove the supernatant, the sediment was resuspended in JC-1 staining buffer. Finally, samples were analyzed using a multifunctional microplate reader with excitation wavelengths of 525/488 nm and emission wavelengths of 590/525 nm.

### 4.7. Statistical Analysis

Data analysis was conducted using SPSS 26.0 software (Chicago, IL, USA). The Shapiro–Wilk test was employed to evaluate the normality of the data distribution. Afterwards, assuming the data showed a normal distribution, a one-way ANOVA LSD test was conducted to assess variations in these parameters. The significance level was established at a *p*-value of less than 0.05, unless stated otherwise. The results are displayed using the mean ± standard deviation format.

### 4.8. Extraction of Metabolites

The sample (Con group: fresh semen; the diluent group was preserved for 3 days; in the 3-day Cur group, 25 μmol/L of curcumin was added to the semen and preserved for 3 days) was removed from a −80 °C refrigerator and thawed on ice. A total of 100 μL of the sample was then transferred to an EP tube, to which 400 μL of extraction solution (methanol: acetonitrile = 1:1, *v*/*v*, with isotopically labeled internal standard mixture) was added. The mixture was then agitated for 30 s, followed by sonication for 10 min in an ice bath. After that, the sample was kept at −40 °C for 1 h before being centrifuged at 4 °C and at a speed of 12,000 rpm for 15 min. The supernatant was collected into an injection bottle for machine testing.

### 4.9. LC-MS Detection of the Metabolome

The analytical instrument utilized for this test was the LC-MS instrument analysis platform Vanquish (UPLC, Thermo Fisher Scientific, Waltham, MA, USA; Orbitrap Exploris 120, Thermo Fisher Scientific) with an ACQUITY UPLC BEH Amide column (1.7 μm, 2.1 mm × 50 mm). The injection volume was 2 μL and the autosampler temperature was set at 4 °C. Mobile phase A consisted of 25 mmol/L H_2_O + 25 mmol/L HCOOH, while mobile phase B comprised 100% CAN. Subsequently, mass spectrum results were collected using Xcalibur software (version: 4.4, Thermo) with the following parameters: sheath gas flow rate: 50 Arb; aux gas flow rate: 15 Arb; capillary temperature: 320 °C; full ms resolution: 60,000; MS/MS resolution: 15,000; collision energy: SNCE20/30/40; spray voltage: 3.8 kV (positive) or −3.4 kV (negative).

### 4.10. Metabolome Data Processing and Analysis

The mass spectrometry raw data were initially converted into mzXML format using ProteoWizard software (version 3.0.6150) for subsequent analysis. XCMS was utilized for retention time correction, peak identification, peak extraction, peak integration, and peak alignment. Both the R program package and the self-built secondary mass spectrometry database (AllwegeneDB) were used for data analysis.

For the processed data, multivariate statistical analyses and univariate analyses were performed. Orthogonal partial least squares discriminant analysis (OPLS-DA) was employed to differentiate overall differences in metabolic profiles between groups, particularly differential metabolites with Cur treatment. OPLS-DA models the relationship between predictor variables (X matrix) and response variable (Y matrix), enabling identification of the main differences between groups in multivariable space. To avoid overfitting of models, 200 response rank tests (RPTs) were conducted to check model quality. This validation approach aided in determining model stability and predictive performance.

Variable importance projection (VIP) scores obtained from the OPLS-DA model were utilized to assess the impact strength and explanatory power of each metabolite on sample classification or discrimination, as well as to identify biologically meaningful differential metabolites. Additionally, differences in metabolites were compared using a t-test and fold-change analysis. Significantly different metabolites were selected based on a combination of statistically significant VIP values and *p* values for a normalized peak area two-tailed Student’s *t*-test (i.e., *p* values < 0.05, fold change >1.5 or <0.67), with VIP greater than 1 indicating significance. These identified metabolites were considered to differ significantly between conditions or groups. For the selected significantly different metabolites, metabolic pathway enrichment analysis based on the Kyoto Encyclopedia of Genes and Genomes (KEGG) database was performed. The pathways could be further screened using enrichment analysis and topological analysis to identify key pathways with the highest correlation with metabolite differences. Finally, SPSS 26.0 software (Chicago, IL, USA) was used to explore associations among the different variables in the data set.

### 4.11. Lipid Extraction

The method was as follows: Take appropriate samples and add 200 μL water. Mix by vortexing and add 800 μL methyl tert-butyl ether (MTBE), then vortex again. Add 240 μL pre-cooled methanol and vortex. Ultrasonicate in a cold water bath for 20 min, then centrifuge at 14,000× *g* for 15 min at 10 °C. Collect the upper organic phase, dry with nitrogen, add 200 μL of 90% isopropyl alcohol/acetonitrile, and vortex. Centrifuge 90 μL of the complex solution at 14,000× *g* for 15 min at 10 °C and analyze the supernatant.

### 4.12. Lipid Group LC-MS Detection

Samples were separated using the UHPLC Nexera LC-30A ULHPLC system with a C18 chromatographic column at a temperature of 45 °C and a flow rate of 300 μL/min. The mobile phase composition included the following: A—acetonitrile aqueous solution (acetonitrile: water = 6:4, *v*/*v*) + 0.1% formic acid + 0.1 mM ammonium formate; and B—acetonitrile isopropanol solution (acetonitrile: isopropanol = 1:9, *v*/*v*) + 0.1% formic acid + 0.1 mM ammonium formate. The gradient elution procedure was as follows: from 0 to 2 min, B was maintained at 30%; from 2 to 25 min, B was linearly changed from 30% to 100%; and then B was maintained at 30% from 25 to 35 min. Samples were stored in a −10 °C autosampler throughout the analysis to minimize potential instrument detecting signal fluctuation influences, and continuous sample analysis was conducted in random order. The samples were separated by UHPLC mass spectroscopy using a QExactive series mass spectrometer (Thermo Scientific™). The ESI parameters used were as follows: heater temp: 300 °C; sheath gas flow rate: 45 arb; aux gas flow rate: 15 arb; sweep gas flow ratel arb; spray voltage: 3.0 KV; capillary temp: 350 °C; S-lens RF level: 50%; MS1 scan range: 200–1800.

### 4.13. Lipidome Data Processing and Analysis

Peak recognition, peak extraction, lipid identification, and secondary identification of lipid molecules were performed using LipidSearch. The main parameters used were as follows: precursor tolerance: 5 ppm; product tolerance: 5 ppm; and product ion threshold: 5%. The data obtained from LipidSearch were subjected to univariate statistical analysis, multivariate statistical analysis, and hierarchical clustering analysis. The univariate statistical analysis included Student’s *t*-test, a non-parametric test, and variant multiplicity analysis. Multivariate statistical analysis involved OPLS-DA. Metabolites with VIP scores in the projection that had either FC > 1.5 or FC < 0.67 and a *p* value < 0.05 were selected as class discriminants. Additionally, correlations between different lipids were examined using Pearson’s correlation coefficient with a significance level of *p* < 0.05 indicating a significant correlation. A correlation coefficient tends toward 1 or −1 when the linear relationship between two lipids strengthens; it approaches 1 in a positive correlation and −1 in a negative correlation situation.

## 5. Conclusions

In conclusion, the supplementation of 25 µmol/L Cur during semen preservation notably enhanced the quality of Hainan black goat sperm (including sperm motility, motor performance, plasma membrane integrity, and acrosomal integrity), antioxidant capacity (increased activities of T-AOC, CAT, SOD, etc.), and energy metabolism (levels of MMP and ATP). Decreased levels of ROS and MDA were also observed. Further studies indicated that Cur could regulate metabolites such as malic acid, niacin, leucine, and isoleucine in the seminal plasma of black goats to influence metabolic processes like the citric acid cycle, cholesterol metabolism, fatty acid metabolism, and cAMP signaling pathway, thereby ensuring semen quality. Additionally, the lipid composition of preserved sperm was analyzed, revealing a variety of different lipid subclasses, and significant differences in these lipid molecules were detected among different groups, highlighting variations in lipid metabolism patterns. High levels of PC and TG might play a protective role by maintaining membrane integrity, while lower levels of AcCa suggest the possibility of enhancing antioxidant capacity through glycolytic pathways. Overall, this study offers novel insights into the application of curcumin in the domain of semen preservation and screens out potential biomarkers during the preservation process.

## Figures and Tables

**Figure 1 ijms-25-10200-f001:**
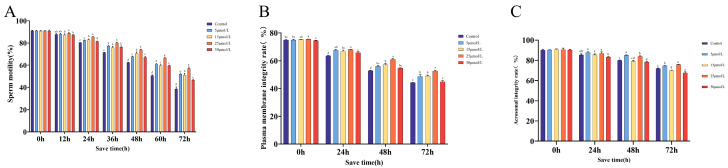
Effects of curcumin on sperm quality at different time points (*n* = 10). (**A**) Impact of curcumin on sperm viability rate. (**B**) Influence of curcumin on plasma membrane integrity. (**C**) Effect of curcumin on acrosome integrity. In the same group, different lowercase letters were found to be statistically significant (*p* < 0.05), but the same letter means no significant difference (*p* > 0.05).

**Figure 2 ijms-25-10200-f002:**
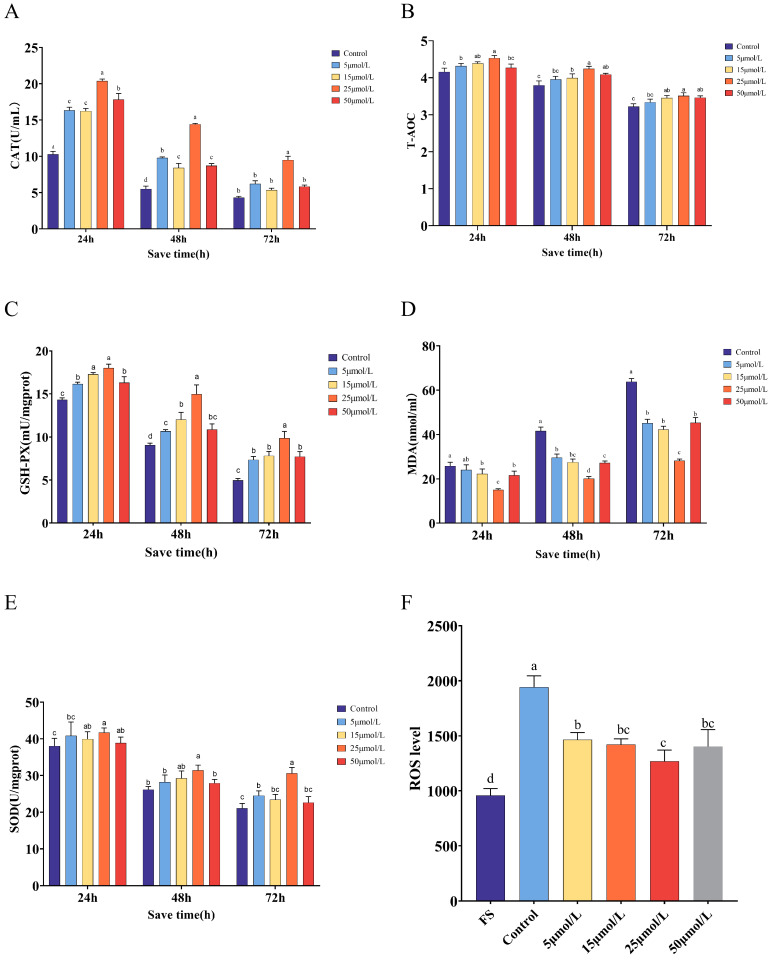
Effects of curcumin on antioxidant capacity at different time points. (**A**) The effect of curcumin on the sperm CAT. (**B**) Effect of curcumin on sperm T-AOC. (**C**) The effect of curcumin on sperm GSH-PX. (**D**) The effect of curcumin on the sperm MDA. (**E**) The effect of curcumin on sperm SOD. (**F**) The effect of curcumin on sperm ROS. In the same group, different lowercase letters were significant (*p* < 0.05) and the same letter means no significant difference (*p* > 0.05).

**Figure 3 ijms-25-10200-f003:**
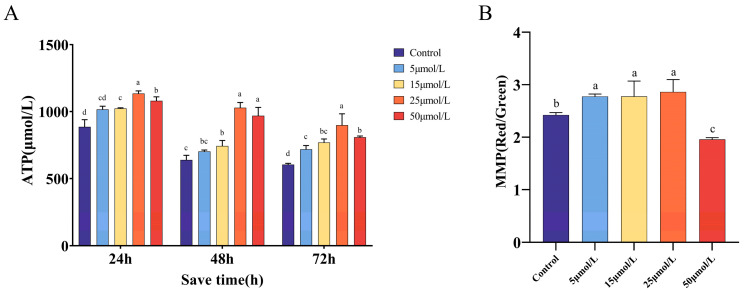
Effects of curcumin on mitochondrial function at different time points. (**A**) Effect of curcumin on sperm ATP content. (**B**) The effect of curcumin on sperm MMP. In the same group, different lowercase letters were found to be significantly different (*p* < 0.05) and the same letter means no significant difference (*p* > 0.05).

**Figure 4 ijms-25-10200-f004:**
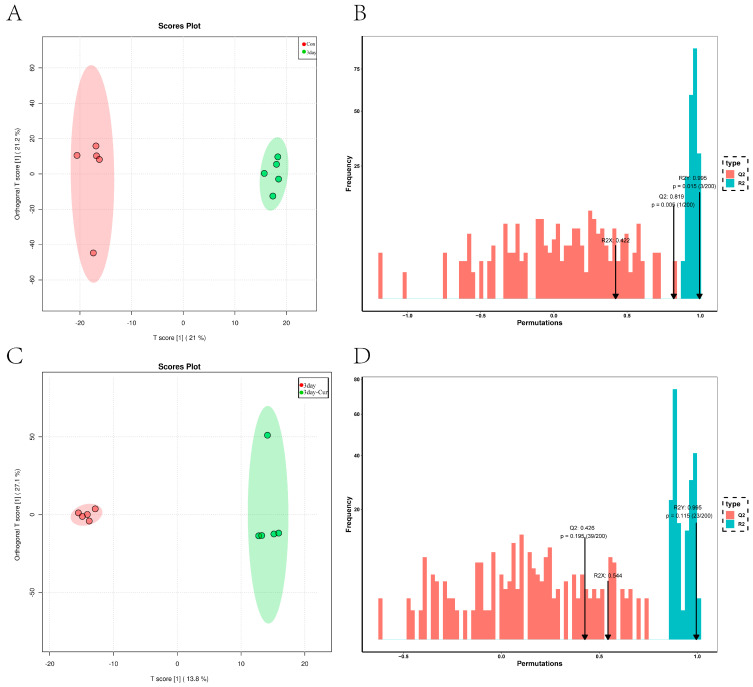
Multivariate statistical analysis. (**A**,**C**) depicts the OPLS-DA score for each group, with the abscissa representing the fraction of the sample on the first principal component and the ordinate representing the fraction of the sample on the second principal component. (**B**,**D**) displays the model validation permutation test diagram, where the abscissa indicates the accuracy of the model and the ordinate represents the frequency of accuracy in 200 permutation tests. The arrow denotes the position of accuracy of the OPLS-DA model. Furthermore, R2X and R2Y denote the interpretation rate of the X and Y matrices, respectively, and Q2 illustrates the predictive ability of the model.

**Figure 5 ijms-25-10200-f005:**
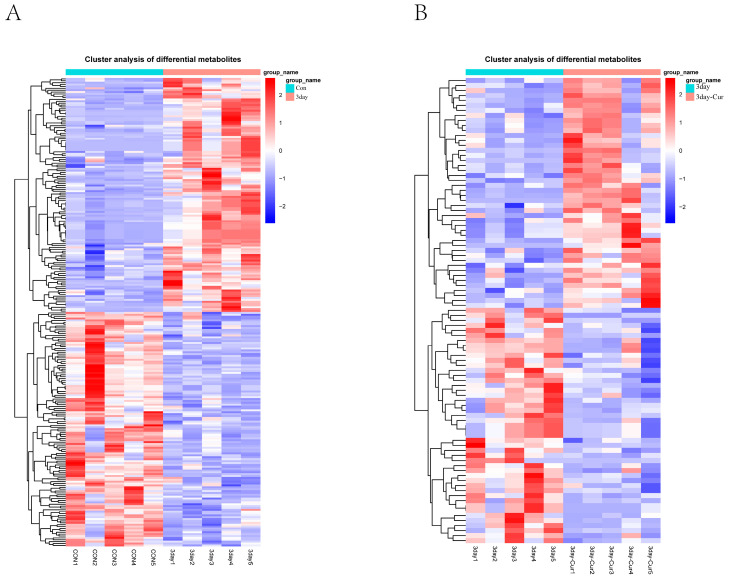
Heat map of the differential metabolites showing relative contents in different colors. (**A**) Heat maps of the differential metabolites in the diluent and control groups were described. (**B**) Heat maps of different metabolites between curcumin group and diluent group. The color intensity indicates the expression level, with red representing higher expression and blue representing lower expression. Columns represent samples, while rows represent metabolite names.

**Figure 6 ijms-25-10200-f006:**
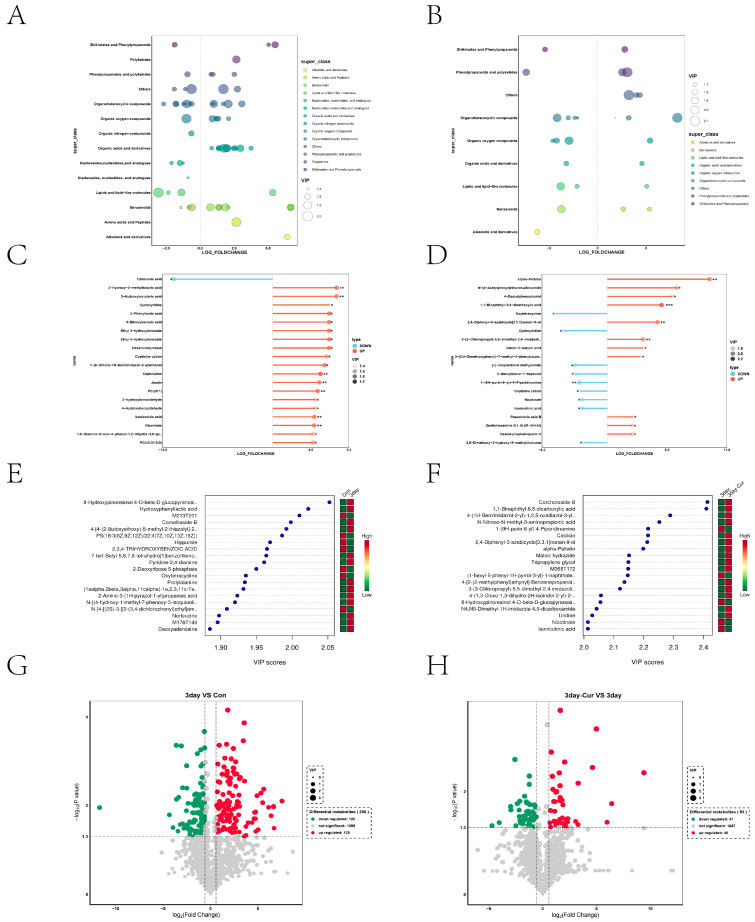
Statistical data on the upregulation and downregulation of different metabolites in each group analyzed. (**A**,**B**) Classification statistics of metabolic differences in each group. (**C**,**D**) Shows the top up- and downregulated substances. (**E**,**F**) The top 20 differentially expressed metabolites were identified using the VIP values. Red indicates higher expression of metabolites in the corresponding group, and green indicates lower expression in the corresponding group. (**G**,**H**) Volcano plot illustrates the up- and downregulated metabolites. “*” indicates 0.01 < *p* < 0.05; “**” indicates *p* < 0.01; “***” indicates *p* < 0.001.

**Figure 7 ijms-25-10200-f007:**
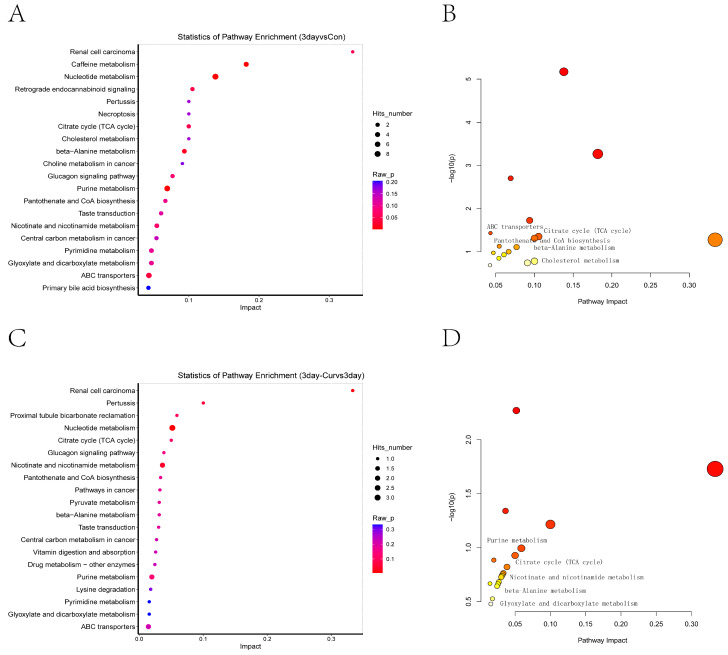
Bubble plots depicting enrichment pathways for all differential metabolites within each group. (**A**,**B**) shows the enrichment pathway in the diluent group, (**C**,**D**) is the Cur group. The number of DAMs in each pathway was counted, and enrichment factors and *p* values were shown.

**Figure 8 ijms-25-10200-f008:**
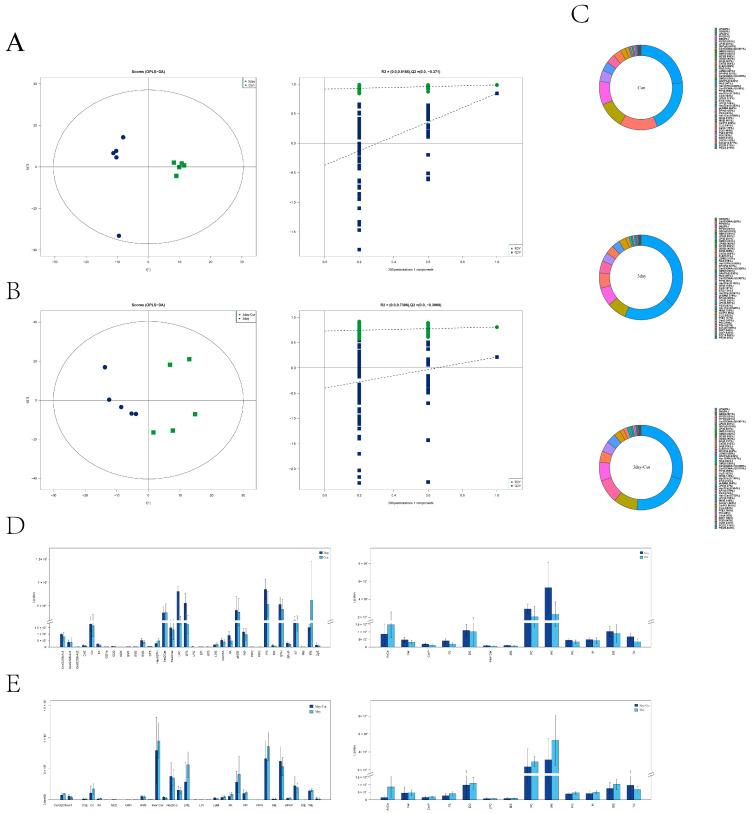
Multivariate statistical analysis. (**A**) represents the diluent group’s OPLS-DA scoring charts and the statistical verification chart. (**B**) represents the Cur group’s OPLS-DA scoring charts and the statistical verification chart. (**C**) The proportion of lipid molecular classification in each group. (**D**) Statistics on the abundance of lipid subclasses in the diluent group. (**E**) Statistics on the abundance of lipid subclasses in the Cur group.

**Figure 9 ijms-25-10200-f009:**
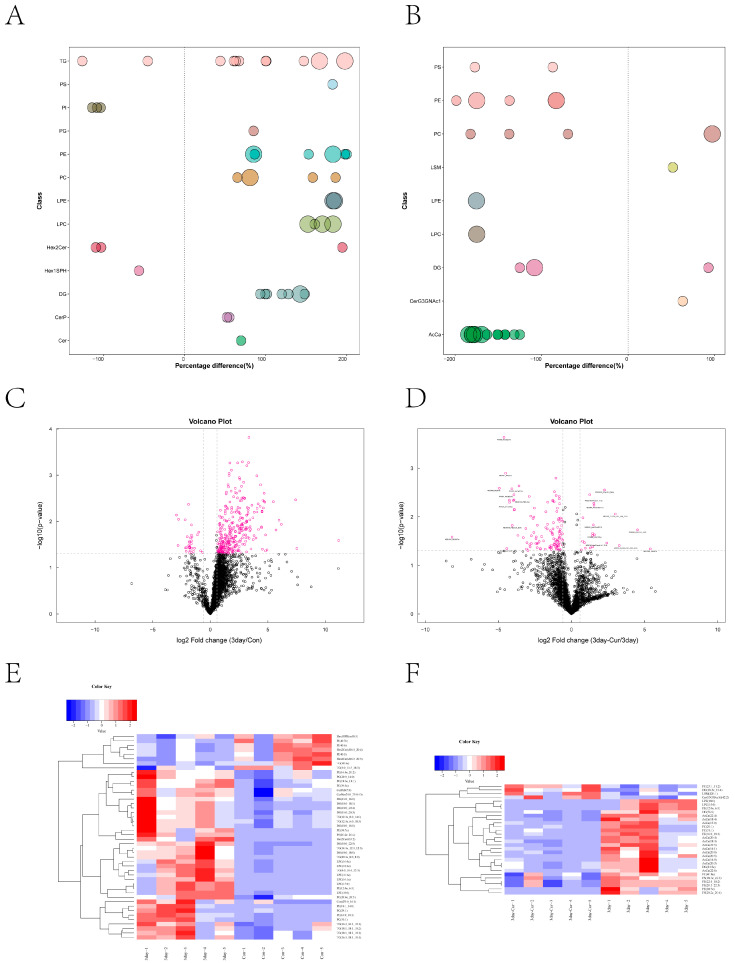
Identification statistics and cluster analysis of potential lipid molecular markers. (**A**,**B**) Bubble map of relative differences in the lipid molecular classification in each group. The horizontal coordinate is the relative difference, the vertical coordinate is the type of lipid, and the point represents the *p*-value of each lipid. The larger the value, the larger the point, and vice versa. (**C**,**D**) A volcano plot with a significant differential distribution of the lipid molecules in each group. Rose dots are significantly differential lipid molecules screened by univariate statistical analysis. (**E**,**F**) Hierarchical clustering results of significantly different lipid molecules in each group. The blue part represents downregulation of expression and the red represents upregulation. Significantly different lipid molecule names are shown on the right side.

**Figure 10 ijms-25-10200-f010:**
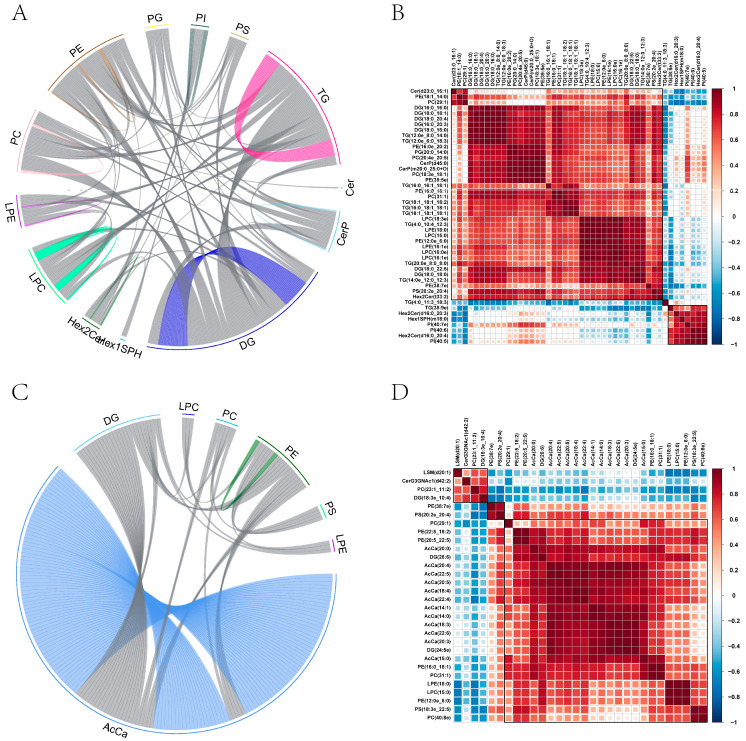
Correlation between differential lipids. (**A**) Paper subclass correlation chord chart in the diluent group. (**B**) Heat map of lipid correlation analysis in the diluent group. (**C**) Paper subclass correlation chord chart in the Cur group. (**D**) Heat map of lipid correlation analysis in the Cur group.

**Table 1 ijms-25-10200-t001:** Effects of different concentrations of curcumin on Hainan black goat ram sperm kinetic parameters stored at 16 °C.

Index	Time of Storage (h)	Control	5 μmol/L	15 μmol/L	25 μmol/L	50 μmol/L
VSL (μm/s)	0	30.63 ± 0.48	30.55 ± 1.20	29.12 ± 0.60	29.53 ± 10.38	30.12 ± 1.23
	24	27.05 ± 1.18	26.42 ± 0.39	26.85 ± 1.99	27.45 ± 1.00	26.27 ± 0.37
	48	21.18 ± 1.23 ^b^	20.82 ± 0.70 ^b^	21.38 ± 0.68 ^ab^	22.77 ± 0.43 ^a^	20.73 ± 0.48 ^b^
	72	11.50 ± 0.75	11.60 ± 1.08	11.25 ± 0.96	11.43 ± 1.62	11.00 ± 1.13
VCL (μm/s)	0	32.07 ± 1.33	32.35 ± 1.42	32.02 ± 1.90	32.26 ± 1.45	31.63 ± 1.49
	24	27.24 ± 0.46 ^a^	25.66 ± 0.89 ^b^	25.63 ± 0.46 ^b^	25.90 ± 0.51 ^b^	25.07 ± 0.41 ^b^
	48	21.45 ± 1.19	20.80 ± 0.73	20.81 ± 0.65	21.63 ± 0.41	20.50 ± 1.09
	72	15.70 ± 0.40 ^ab^	15.96 ± 0.47 ^ab^	16.06 ± 0.94 ^b^	16.66 ± 0.99 ^a^	15.18 ± 0.70 ^a^
VAP (μm/s)	0	46.91 ± 1.36	47.20 ± 1.61	47.11 ± 1.11	47.27 ± 2.48	47.04 ± 1.30
	24	44.75 ± 0.74	45.39 ± 0.96	44.53 ± 0.43	44.75 ± 0.96	45.24 ± 1.01
	48	40.13 ± 0.61	39.72 ± 1.21	39.76 ± 0.56	40.24 ± 0.61	39.43 ± 0.77
	72	37.57 ± 0.67 ^ab^	37.48 ± 0.93 ^ab^	36.72 ± 0.48 ^b^	38.17 ± 0.31 ^a^	36.43 ± 0.85 ^b^

a, b denoted within the same line with different superscripts indicate significantly differences at *p* < 0.05.

**Table 2 ijms-25-10200-t002:** Differential lipid species in the diluent group.

Lipid Ion	Lipid Group	Ion Formula	FC	*p*-Value	VIP
PE(18:1_14:0) - H	PE(32:1) - H	C37 H71 O8 N1 P1	183.51	0.038	2.12
PE(16:0_18:1) - H	PE(34:1) - H	C39 H75 O8 N1 P1	7.49	0.019	1.34
PE(38:7e) - H	PE(38:7e) - H	C43 H73 O7 N1 P1	2262.42	0.026	8.03
PS(20:2e_20:4) - H	PS(40:6e) - H	C46 H79 O9 N1 P1	21.89	0.014	1.27
Hex2Cer(d16:0_20:3) + HCOO	Hex2Cer(d36:3) + HCOO	C49 H88 O15 N1	0.29	0.036	1.09
Hex2Cer(d16:0_20:4) + HCOO	Hex2Cer(d36:4) + HCOO	C49 H86 O15 N1	0.32	0.030	2.59
Hex2Cer(t33:2) + HCOO	Hex2Cer(t33:2) + HCOO	C46 H84 O16 N1	72.23	0.012	1.36
LPC(16:0e) + HCOO	LPC(16:0e) + HCOO	C25 H53 O8 N1 P1	7.35	0.003	1.82
LPC(16:1e) + HCOO	LPC(16:1e) + HCOO	C25 H51 O8 N1 P1	9.11	0.010	3.33
LPE(16:1e) - H	LPE(16:1e) - H	C21 H43 O6 N1 P1	22.55	0.006	2.32
LPE(18:0) - H	LPE(18:0) - H	C23 H47 O7 N1 P1	25.48	0.004	1.25
DG(18:0_16:0) + NH4	DG(34:0) + NH4	C37 H76 O5 N1	2.97	0.028	1.93
DG(18:0_18:0) + NH4	DG(36:0) + NH4	C39 H80 O5 N1	6.70	0.011	3.16
DG(18:0_18:1) + NH4	DG(36:1) + NH4	C39 H78 O5 N1	3.98	0.023	1.55
DG(16:0_20:3) + H	DG(36:3) + H	C39 H71 O5	3.05	0.020	1.93
DG(18:0_20:4) + H	DG(38:4) + H	C41 H73 O5	4.55	0.016	1.69
DG(18:0_22:5) + NH4	DG(40:5) + NH4	C43 H78 O5 N1	5.96	0.001	1.05
Hex1SPH(m18:0) + H - H2O	Hex1SPH(m18:0) + H - H2O	C24 H48 O5 N1	0.56	0.043	2.00
LPC(15:0) + H	LPC(15:0) + H	C23 H49 O7 N1 P1	22.55	0.006	1.50
LPC(18:3e) + H	LPC(18:3e) + H	C26 H51 O6 N1 P1	12.27	0.004	3.04
PC(29:1) + H	PC(29:1) + H	C37 H73 O8 N1 P1	27.90	0.043	1.90
PC(31:1) + H	PC(31:1) + H	C39 H77 O8 N1 P1	8.52	0.023	1.92
PC(18:3e_18:1) + H	PC(36:4e) + H	C44 H83 O7 N1 P1	2.35	0.009	1.64
PC(20:4e_20:5) + H	PC(40:9e) + H	C48 H81 O7 N1 P1	1.96	0.038	1.45
PE(12:0e_6:0) + H	PE(18:0e) + H	C23 H49 O7 N1 P1	22.58	0.006	1.49
PE(16:0e_20:2) + Na	PE(36:2e) + Na	C41 H80 O7 N1 P1 Na1	2.52	0.033	1.55
PE(39:5e) + H	PE(39:5e) + H	C44 H81 O7 N1 P1	2.48	0.006	4.78
Cer(d23:0_16:1) + H	Cer(d39:1) + H	C39 H78 O3 N1	2.06	0.043	1.36
PG(20:0_14:0) + H	PG(34:0) + H	C40 H80 O10 N0 P1	2.48	0.036	2.92
PI(40:5) + NH4	PI(40:5) + NH4	C49 H89 O13 N1 P1	0.32	0.047	3.90
PI(40:6) + NH4	PI(40:6) + NH4	C49 H87 O13 N1 P1	0.27	0.033	1.06
PI(40:7e) + NH4	PI(40:7e) + NH4	C49 H87 O12 N1 P1	0.30	0.043	2.69
TG(4:0_10:4_12:3) + NH4	TG(26:7) + NH4	C29 H44 O6 N1	10.85	0.009	4.93
TG(4:0_11:3_18:3) + Na	TG(33:6) + Na	C36 H56 O6 Na1	0.63	0.047	1.15
TG(12:0e_8:0_14:0) + Na	TG(34:0e) + Na	C37 H72 O5 Na1	3.02	0.029	1.82
TG(20:0e_8:0_8:0) + Na	TG(36:0e) + Na	C39 H76 O5 Na1	170.56	0.003	3.58
TG(12:0e_6:0_18:3) + H	TG(36:3e) + H	C39 H71 O5	3.00	0.028	1.82
TG(14:0e_12:0_12:3) + H	TG(38:3e) + H	C41 H75 O5	6.54	0.012	2.87
TG(38:9e) + Na	TG(38:9e) + Na	C41 H62 O5 Na1	0.23	0.037	1.37
TG(16:0_16:1_18:1) + NH4	TG(50:2) + NH4	C53 H102 O6 N1	1.56	0.047	2.09
TG(16:0_18:1_18:1) + NH4	TG(52:2) + NH4	C55 H106 O6 N1	2.01	0.029	1.58
TG(18:1_18:1_18:1) + NH4	TG(54:3) + NH4	C57 H108 O6 N1	1.92	0.050	1.20
TG(18:1_18:1_18:2) + NH4	TG(54:4) + NH4	C57 H106 O6 N1	1.87	0.040	1.11
CerP(d45:0) + H	CerP(d45:0) + H	C45 H93 O6 N1 P1	1.70	0.047	2.07
CerP(m20:0_25:0+O) + H	CerP(m45:0+O) + H	C45 H93 O6 N1 P1	1.76	0.040	2.23
DG(16:0_16:0) + NH4	DG(32:0) + NH4	C35 H72 O5 N1	2.74	0.050	2.67

**Table 3 ijms-25-10200-t003:** Differential lipid species in the Cur group.

Lipid Ion	Lipid Group	Ion Formula	FC	*p*-Value	VIP
PE(16:0_18:1) - H	PE(34:1) - H	C39 H75 O8 N1 P1	0.19	0.028	1.34
PE(38:7e) - H	PE(38:7e) - H	C43 H73 O7 N1 P1	0.00	0.026	8.73
PS(20:2e_20:4) - H	PS(40:6e) - H	C46 H79 O9 N1 P1	0.06	0.015	1.34
PS(18:3e_22:5) - H	PS(40:8e) - H	C46 H75 O9 N1 P1	0.39	0.042	1.36
LPE(18:0) - H	LPE(18:0) - H	C23 H47 O7 N1 P1	0.07	0.004	1.36
AcCa(14:1) + H	AcCa(14:1) + H	C21 H40 O4 N1	0.14	0.037	1.47
AcCa(15:0) + H	AcCa(15:0) + H	C22 H44 O4 N1	0.17	0.021	1.06
LPC(15:0) + H	LPC(15:0) + H	C23 H49 O7 N1 P1	0.07	0.007	1.64
LSM(d20:1) + H	LSM(d20:1) + H	C25 H54 O5 N2 P1	1.70	0.032	1.01
PC(29:1) + H	PC(29:1) + H	C37 H73 O8 N1 P1	0.05	0.045	3.23
PC(31:1) + H	PC(31:1) + H	C39 H77 O8 N1 P1	0.19	0.035	1.89
PC(23:1_11:2) + H	PC(34:3) + H	C42 H79 O8 N1 P1	2.90	0.005	2.15
PC(40:8e) + H	PC(40:8e) + H	C48 H83 O7 N1 P1	0.48	0.038	5.30
PE(12:0e_6:0) + H	PE(18:0e) + H	C23 H49 O7 N1 P1	0.07	0.007	1.63
PE(22:5_18:2) + Na	PE(40:7) + Na	C45 H76 O8 N1 P1 Na1	0.41	0.005	1.41
PE(20:5_22:5) + H	PE(42:10) + H	C47 H75 O8 N1 P1	0.41	0.005	1.40
AcCa(18:3) + H	AcCa(18:3) + H	C25 H44 O4 N1	0.17	0.010	1.29
AcCa(18:4) + H	AcCa(18:4) + H	C25 H42 O4 N1	0.08	0.002	1.19
AcCa(20:0) + H	AcCa(20:0) + H	C27 H54 O4 N1	0.21	0.035	1.64
AcCa(20:3) + H	AcCa(20:3) + H	C27 H48 O4 N1	0.23	0.046	2.24
AcCa(20:4) + H	AcCa(20:4) + H	C27 H46 O4 N1	0.05	0.005	6.09
AcCa(20:5) + H	AcCa(20:5) + H	C27 H44 O4 N1	0.05	0.005	6.11
CerG3GNAc1(d42:2) + H	CerG3GNAc1(d42:2) + H	C68 H125 O23 N2	1.92	0.044	1.01
AcCa(22:4) + H	AcCa(22:4) + H	C29 H50 O4 N1	0.04	0.000	2.21
AcCa(22:5) + H	AcCa(22:5) + H	C29 H48 O4 N1	0.06	0.003	8.41
AcCa(22:6) + H	AcCa(22:6) + H	C29 H46 O4 N1	0.14	0.042	7.85
AcCa(14:0) + H	AcCa(14:0) + H	C21 H42 O4 N1	0.10	0.027	7.06
DG(24:5e) + NH4	DG(24:5e) + NH4	C27 H48 O4 N1	0.23	0.045	2.24
DG(26:6) + NH4	DG(26:6) + NH4	C29 H48 O5 N1	0.30	0.008	1.14
DG(18:3e_10:4) + NH4	DG(28:7e) + NH4	C31 H52 O4 N1	2.73	0.041	1.29

## Data Availability

Data are contained within the article and Appendix A.

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
