# Peer review of "Exploratory Metabolomics and Lipidomics Profiling Contributes to Understanding How Curcumin Improves Quality of Goat Semen Stored at 16 °C in Tropical Areas"

_ijms, 2024, doi:10.3390/ijms251810200_

Round 1
Reviewer 1 Report
Comments and Suggestions for Authors
This study takes on an ever-interesting theme in animal reproduction science (improvement of the procedures that are currently established to preserve animal spermatozoa at low temperatures) with a very intriguing twist – by employing metabolomics and lipidomics which are becoming more and more popular in molecular andrology.
The study is very complex, comprising a wide array of methodologies, and providing interesting data.
Nevertheless, the manuscript needs to address several issues, particularly in the methodology section which are either lacking or confusing.
Introduction
- The section seems very generic. It would benefit from specific related to goat semen preservation – how does it differ from other domestic or farm animals? Why is liquid storage more preferred in comparison to cryopreservation? What are the peculiarities of goat spermatozoa?
- The aim needs to be reformulated. From its current version it seems that this was an in vivo study. The main and partial goals should be clearly defined.
Methodology
- First of all, most of the chemicals used lack a manufacturer and location. Second, the equipment is not described. What CASA system was used? What microscopes and plate readers were used? Manufacturers and locations are needed as well.
- The rationale behind the concentration range should be explained. Was curcumin diluted directly in the medium or pre-diluted in a vehicle (DMSO for example)?
- What served as the control?
- The time intervals of analysis should be explained.
- In what type of sample were the oxidative and antioxidant markers determined? Diluted semen? Sperm lysates?
- The sample processing for the extraction of metabolites is where I got seriously confused. In lines 644/645 it is stated that “The sample (Con Group: fresh semen; the 3day group was preserved for 3 days. In the 3day-Cur group , 25μmol/L of curcumin was added to the semen and preserved for 3 days) was removed from a -80°C refrigerator and thawed on ice”. So, when did the samples end up from 16°C to -80°C ? How many samples were processed for this purpouse?
- What is MTBE (Line 702)?
- What served as negative control in the -omic experiments? In other words, did not the medium interfere with the results?
Discussion
- The section is quite vast, which is expected given the amount of the data collected. Nevertheless, at some point the discussion turns into a description of each lipid class studied in the experiments and their physiological roles. Except for triglycerides, is not clear how curcumin supplementation affected their dynamics within the experiment.
- Limitations of the study need to be properly discussed.
Conclusion
- This section is very vague and essentially directed to the effects of curcumin on the sperm quality per se, which is not particularly novel given the evidence collected from previous studies. The real novelty based on the interactions of curcumin with the lipids is not summarized.
Comments on the Quality of English LanguageThere are numerous typos throughout the paper, which need to be corrected. The sections 4.3.2. Integrity of the plasma membrane and 4.8. LC-MS detection of the metabolome are written in present tense. The section 4.10. Lipid extraction is literally written as a protocol, not as a narrative.
Author Response
Introduction
The section seems very generic. It would benefit from specific related to goat semen preservation – how does it differ from other domestic or farm animals? Why is liquid storage more preferred in comparison to cryopreservation? What are the peculiarities of goat spermatozoa?
The aim needs to be reformulated. From its current version it seems that this was an in vivo study. The main and partial goals should be clearly defined.
A:We chose Hainan black goat as the research object, Hainan black goat as Hainan native tropical goat, is one of the endemic species in Hainan region. Long-term regional climatic selection has shaped this endemic variety. Hainan black goat is also one of the mutton sheep breeds in the hot area of southern China, and small-scale grazing is mostly used as the main breeding mode. During the breeding season, the preservation of frozen semen and its use is limited by the ambient temperature and the technical level is not commonly used, so the use of easy liquid dilution and preservation at room temperature are much more common. However, we refer to relevant literature and found that in semen preservation studies, goat-related studies are very limited and there are large gaps. In the research on the preservation of goat semen, it is basically about low temperature and cryopreservation. This also suggests that there are many limitations in the direction of body fluid preservation. We believe that compared with low temperature or frozen storage does not require expensive equipment and is easy to operate, it has a greater advantage for short-term storage. At the same time, there are more polyunsaturated fatty acids in goat sperm, and the damage caused by low temperature and freezing will aggravate the process of lipid peroxidation. Therefore, in order to meet our short-term storage or short-distance transportation conditions, we focused our research on body fluid storage.
The aim of this study was to investigate whether Curcumin can enhance sperm antioxidant capacity and reduce levels of reactive oxygen species and malondialdehyde content in Hainan Black goat. In addition, significant differences in metabolomics and lipidomics characteristics were compared among semen samples from different groups to elucidate the metabolic pathways and key markers that may contribute to these differences. The results of this study will facilitate a deeper understanding of the molecular mechanisms underlying black goat sperm fluid preservation capacity. Furthermore, it will provide relevant information and guidance for optimizing semen preservation techniques and quality assessment methods.
Methodology
First of all, most of the chemicals used lack a manufacturer and location. Second, the equipment is not described. What CASA system was used? What microscopes and plate readers were used? Manufacturers and locations are needed as well.
A:Following your recommendation, we checked the information on the relevant chemicals used. From 4.1 to 4.4, we show the manufacturer and site used to prepare the diluent and antioxidant test kit. But due to our negligence, we missed the relevant information about the devices used for sperm quality detection. We have supplemented this in the manuscript.
The rationale behind the concentration range should be explained. Was curcumin diluted directly in the medium or pre-diluted in a vehicle (DMSO for example)?
A:In this experiment, we used pre-diluted dissolution in DMSO, and we have added the manuscript.
What served as the control?
A:The control variable for this experiment depends on whether curcumin was added to the diluent, so the control group was added only to the diluent.
The time intervals of analysis should be explained.
A:Thank you for the reminder. Prior to this trial, we performed a pretest, and we found that the semen stored after the addition of only the dilutions reduced the sperm viability to below 60% after 72h. We believe that the AI of sperm AI under liquid preservation conditions. Therefore, the total time we take at 72h is dependent on the above reasons. The time selection of the detection indicators was based on each day (24h), with the relevant antioxidant indicators at 24h or 72h. The relevant references are as follows:
Zhang, L.; Wang, Y.; Sohail, T.; Kang, Y.; Niu, H.; Sun, X.; Ji, D.; Li, Y. Effects of Taurine on Sperm Quality during Room Temperature Storage in Hu Sheep. Animals 2021, 11, 2725.
Ji, K.; Wei, J.; Fan, Z.; Zhu, M.; Yuan, X.; Zhang, S.; Li, S.; Xu, H.; Ling, Y. Preservative Effects of Curcumin on Semen of Hu Sheep. Animals 2024, 14, 947.
Zhang X, Hu Z-T, Li Y, Li Y-X, Xian M, Guo S-M, Hu J-H: Effect of Astragalus polysaccharides on the cryopreservation of goat semen. Theriogenology 2022, 193:47-57.
In what type of sample were the oxidative and antioxidant markers determined? Diluted semen? Sperm lysates?
A:We tested the diluted semen.
In lines 644/645 it is stated that “The sample (Con Group: fresh semen; the 3day group was preserved for 3 days. In the 3 day-Cur group, 25μmol/L of curcumin was added to the semen and preserved for 3 days) was removed from a -80°C refrigerator and thawed on ice”. So, when did the samples end up from 16°C to -80°C ?
How many samples were processed for this purpouse?
A:We apologize for the confusion caused by our confusion. The specific operation in our test is as follows: Con group is the fresh semen samples. After dilution, part of the diluted semen were taken out and put it in -80℃ refrigerator for metabolome and lipidome detection. The 3day and 3day-Cur were also placed in the-80℃ refrigerator for metabolome and lipidome detection.5 samples in the metabolome and lipidome.
What is MTBE (Line 702)?
A:We demonstrated that methyl-tert-butyl ether (MTBE) extraction allows faster and cleaner lipid recovery and is well suited for automated shotgun profiling. Because of MTBE's low density, lipid-containing organic phase forms the upper layer during phase separation, which simplifies its collection and minimizes dripping losses.
Matyash V, Liebisch G, Kurzchalia TV, Shevchenko A, Schwudke D. Lipid extraction by methyl-tert-butyl ether for high-throughput lipidomics. J Lipid Res. 2008 May;49(5):1137-46.
At the same time, we have added the full name of MTBE:methyl-tert-butyl ether
What served as negative control in the -omic experiments? In other words, did not the medium interfere with the results?
A:In the test, the diluent taken was Tirs-citric acid, we refer to the relevant literature, and the dissolution media will not affect the omics results. The references are as follows:
Xu B, Wang R, Wang Z, Liu H, Wang Z, Zhang W, Zhang Y, Su R, Liu Z, Liu Y et al: Evaluation of lipidomic change in goat sperm after cryopreservation. Frontiers in veterinary science 2022, 9.
Xu B, Wang Z, Wang R, Song G, Zhang Y, Su R, Liu Y, Li J, Zhang J: Metabolomics analysis of buck semen cryopreserved with trehalose. Frontiers in genetics 2022, 13.
Discussion
The section is quite vast, which is expected given the amount of the data collected. Nevertheless, at some point the discussion turns into a description of each lipid class studied in the experiments and their physiological roles. Except for triglycerides, is not clear how curcumin supplementation affected their dynamics within the experiment.
Limitations of the study need to be properly discussed.
A:We have comprehensively combed and changed the discussion section. The results of the paper are summarized and compared with previous studies. The limitations of the trial results are indicated in some sections. For instance:
Intriguingly, the curcumin-treated group demonstrated superior semen quality following preservation, yet presented lower levels of AcCa compared to the three-day preservation group. We postulate that high-quality semen displays exceptional energy metabolism mechanisms and is more inclined towards the glycolytic pathway, thereby reducing reliance on AcCa. This efficient metabolic route is the key to maintaining sperm motility and motor performance. High-quality semen also exhibits outstanding antioxidant properties and can effectively handle intermediate metabolites, including the oxidation products of AcCa, reducing the potential harm of oxidative stress on sperm and thereby influencing AcCa concentration.
Conclusion
This section is very vague and essentially directed to the effects of curcumin on the sperm quality per se, which is not particularly novel given the evidence collected from previous studies. The real novelty based on the interactions of curcumin with the lipids is not summarized.
A:Thank you for your suggestions. We have modified this information from the conclusion section.
In conclusion, the supplementation of 25 µmol/L Cur during semen preservation notably enhanced the sperm quality (including sperm motility, motor performance, plasma membrane integrity, and acrosomal integrity), antioxidant capacity (increased activities of T-AOC, CAT, and SOD, etc.), and energy metabolism (levels of MMP and ATP) of Hainan Black sheep. Decreased levels of ROS and MDA were also observed. Further studies indicated that Cur could regulate metabolites such as malic acid, niacin, leucine, and isoleucine in the seminal plasma of black goats to influence metabolic processes like the citric acid cycle, cholesterol metabolism, fatty acid metabolism, and cAMP signaling pathway, thereby ensuring semen quality. Simultaneously, the lipid composition of preserved sperm was analyzed, revealing a variety of different lipid subclasses, and significant differences in these lipid molecules were detected among different groups, highlighting variations in lipid metabolism patterns. High levels of PC and TG might play a protective role by maintaining membrane integrity, while lower levels of AcCa suggest the possibility of enhancing antioxidant capacity through glycolytic pathways. Overall, this study offers novel insights into the application of curcumin in the domain of semen preservation and screens out potential biomarkers during the preservation process.
Reviewer 2 Report
Comments and Suggestions for Authors
The present article proposes the investigation of impact of curcumin on the viability of sperm, the integrity of the acrosome, antioxidant enzymes, as well as the profiles of metabolomics and lipidomics. In this study the use of curcumin in goat semen improved the integrity of both the plasma membrane and acrosome and increased the levels of important antioxidant enzymes, specifically Superoxide dismutase (SOD), catalase (CAT), and total antioxidant capacity (T-AOC) and reducing the formation of ROS.
This research work present high quality -omics information because a total of 820 metabolites were found being, lipids and lipid molecules the most prevalent groups. The lipidomics study identified a total of 49 lipid subclasses and 2,495 individual lipid molecules. The differential metabolite analysis identified 250 and 93 unique metabolites in each of the different groups, respectively.
Therefore, under my point of view, this article offers a new method to enhance semen preservation at room temperature (16℃) by using of curcumin. This innovative solutions presents new opportunities for enhancing the storage conditions of Hainan black goat semen and emphasizes the possibility of curcumin as a valuable addition to semen preservation tactics. However, further research would be necessary in order to broaden the field of application and for this new workflow to be used as a routine analytical tool.
In general I consider that, this article can be accepted for publication in International Journal of Molecular Sciences, but I have some doubts and curiosities that perhaps the authors could take into account in order to improve the article.
The following are my comments on some general aspects of the work and other more specific suggestions:
1. In the introduction it comments the importance of being able to work at room temperature, which is normally set at 25 ºC or 21 ºC in more specific cases. The authors indicate that the temperature proposed in the study, 16 ºC, is the ambient temperature for tropical climates, but it is not very clear to me if this is a normal working temperature in laboratories or if it is necessary to work in cold rooms conditioned to the study temperature.
Have other temperatures closer to the room temperatures (25 or 21 º C) been tested?
2. This work provides a wealth of information and data that can be very valuable for future research on the use of antioxidants to improve sperm viability in animals and even in humans. The results are adequately presented and discussed, but from my point of view there is an excess of references are included in discussion that do not add value to the research performed. The work, without being a “review paper”, cites almost a hundred articles, some of them without much relation to the proposed work and mostly of them are more than 5-10 years old. For example:
- References 4-12 ¿are all of then necessaries?
- References 30, 31 y 40 ¿are a mistake?
Authors should cite fewer and more current papers.
3. Since there are previous works in which curcumin has been used for semen preservation at different temperatures (references 16-19, 38) it would be necessary to present, in the introduction or discussion, the results obtained.
One of the most innovative and recent tools is the use of nanotechnology to improve the properties and release of antioxidant compounds in, for example, semen samples, through nanoparticles. The authors refer to a single article from the year 2020:
19. Effects of mint, thyme, and curcumin extract nanoformulations on the sperm quality, apoptosis, chromatin decondensation, enzyme activity, and oxidative status of cryopreserved goat semen. Ismail AA, Abdel-Khalek AE, Khalil WA, Yousif AI, Saadeldin IM, Abomughaid MM, El-Harairy MA: Cryobiology 2020, 97:144-152.
Are there no further references to it? It would be interesting to discuss this aspect.
4. In the Results section, in 2.1 sub-section, in addition to studying the proposed kinetic parameters, total and progressive motility are also evaluated. Do the authors have studies and results on these parameters?
- There are many abbreviations that are commonly used throughout the article. Some of them, it is not clear or indicated in the text to which they correspond and some of them do not appear in the list of abbreviations (ALH?, Line 118; Table 1 subscript a) and b) ?) could the authors indicate the meaning of the initials in the text or in legend tables?
5. In the sub-section 2.2 it measures viability (% of sperm moving), membrane integrity (for lipid peroxidation) and acrosome status, but do not measure % DNA fragmentation and DNA strand stability – why not?
6. The conclusions are really poor given the amount of information provided in the paper and the good results obtained.
Author Response
1.In the introduction it comments the importance of being able to work at room temperature, which is normally set at 25 ºC or 21 ºC in more specific cases. The authors indicate that the temperature proposed in the study, 16 ºC, is the ambient temperature for tropical climates, but it is not very clear to me if this is a normal working temperature in laboratories or if it is necessary to work in cold rooms conditioned to the study temperature.
Have other temperatures closer to the room temperatures (25 or 21 º C) been tested?
A:The ambient temperature we selected in this experiment was 16℃, which was the temperature we set in the incubation refrigerator at constant temperature. Our consideration is that we are in a hot climate, in a tropical region, and at a room temperature of over 30℃. Hainan black goat is also one of the mutton sheep breeds in the hot area of southern China, and small-scale grazing is mostly used as the main breeding mode. During the breeding season, the preservation of frozen semen and its use is limited by the ambient temperature and the technical level is not commonly used, so the use of easy liquid dilution and preservation at room temperature are much more common. Related studies have demonstrated that storage temperatures of> 15℃ are feasible, while causing less cold damage to sperm. The relevant studies are conducted as follows:
Li Q, Shaoyong W, Li Y, Chen M, Hu Y, Liu B, Yang G, Hu J. Effects of oligomeric proanthocyanidins on quality of boar semen during liquid preservation at 17 °C. Anim Reprod Sci. 2018 Nov;198:47-56.
Li D, Zhang W, Tian X, He Y, Xiao Z, Zhao X, Fan L, Du R, Yang G, Yu T. Hydroxytyrosol effectively improves the quality of pig sperm at 17 °C. Theriogenology. 2022 Jan 1;177:172-182.
Martín-Hidalgo D, Barón FJ, Bragado MJ, Carmona P, Robina A, García-Marín LJ, Gil MC. The effect of melatonin on the quality of extended boar semen after long-term storage at 17 °C. Theriogenology. 2011 May;75(8):1550-60.
We have not yet followed the room temperature setting in other tests (25 or 21℃), and maybe we can conduct more temperatures later to determine a more suitable storage temperature.
2.This work provides a wealth of information and data that can be very valuable for future research on the use of antioxidants to improve sperm viability in animals and even in humans. The results are adequately presented and discussed, but from my point of view there is an excess of references are included in discussion that do not add value to the research performed. The work, without being a “review paper”, cites almost a hundred articles, some of them without much relation to the proposed work and mostly of them are more than 5-10 years old. For example:
-References 4-12 ¿are all of then necessaries?
A:Thank you for your suggestions. We examined the references of 4-12, and we found that some of the literature was related to the trials we retained, not closely related to the subject.-References 30, 31 y 40 ¿are a mistake?
A:Thank you for your suggestions. We found that the literature 30,31 was improperly cited. We as well as remove the relevant literature. In Literature 40, we found that it did not better explain what we meant, and we replaced the literature.
Authors should cite fewer and more current papers.
3.Since there are previous works in which curcumin has been used for semen preservation at different temperatures (references 16-19, 38) it would be necessary to present, in the introduction or discussion, the results obtained.
One of the most innovative and recent tools is the use of nanotechnology to improve the properties and release of antioxidant compounds in, for example, semen samples, through nanoparticles. The authors refer to a single article from the year 2020:
19.Effects of mint, thyme, and curcumin extract nanoformulations on the sperm quality, apoptosis, chromatin decondensation, enzyme activity, and oxidative status of cryopreserved goat semen. Ismail AA, Abdel-Khalek AE, Khalil WA, Yousif AI, Saadeldin IM, Abomughaid MM, El-Harairy MA: Cryobiology 2020, 97:144-152.
Are there no further references to it? It would be interesting to discuss this aspect.
A:Thank you for your advice. We also found that the literature you listed is the current hot research. It is undeniable that NF is being paid attention to to avoid the accumulation of nanomaterials in the body and improve the solubility and stability of bioactive compounds. We also summarize the novel aspects of this study in the discussion section. Lines 427-437.
4.In the Results section, in 2.1 sub-section, in addition to studying the proposed kinetic parameters, total and progressive motility are also evaluated. Do the authors have studies and results on these parameters?
A:Thank you for your question. We did not focus on the total and progressive motility you mentioned in this study. We are also conducting subsequent trials, and we will pay attention to these parameters during the trial according to your recommendations.
-There are many abbreviations that are commonly used throughout the article. Some of them, it is not clear or indicated in the text to which they correspond and some of them do not appear in the list of abbreviations (ALH? , Line 118; Table 1 subscript a) and b) ?) could the authors indicate the meaning of the initials in the text or in legend tables?
A:Thank you for your reminder. We examined the appearing ALH in 118 lines, which we found was due to our negligence that we wanted to express the VSL and has been corrected in the text.
And what you mentioned about a and b in Table 1, we want to express representations with significant differences in the same column. We have supplemented the corresponding interpretation below Table 1.
5.In the sub-section 2.2 it measures viability (% of sperm moving), membrane integrity (for lipid peroxidation) and acrosome status, but do not measure % DNA fragmentation and DNA strand stability – why not?
A:Thank you for your advice. We focused on the protective effect of curcumin (ensuring semen quality) in this study. We also prepare for follow-up trials to demonstrate the antioxidant capacity of curcumin. In subsequent trials we will likely perform the% DNA fragmentation and DNA strand stability. We felt that this was a very good proposal.
- The conclusions are really poor given the amount of information provided in the paper and the good results obtained.
A:Thank you for your suggestions. We have revised the conclusions based on the revisions of the overall manuscript.
In conclusion, the supplementation of 25 µmol/L Cur during semen preservation notably enhanced the sperm quality (including sperm motility, motor performance, plasma membrane integrity, and acrosomal integrity), antioxidant capacity (increased activities of T-AOC, CAT, and SOD, etc.), and energy metabolism (levels of MMP and ATP) of Hainan Black sheep. Decreased levels of ROS and MDA were also observed. Further studies indicated that Cur could regulate metabolites such as malic acid, niacin, leucine, and isoleucine in the seminal plasma of black goats to influence metabolic processes like the citric acid cycle, cholesterol metabolism, fatty acid metabolism, and cAMP signaling pathway, thereby ensuring semen quality. Simultaneously, the lipid composition of preserved sperm was analyzed, revealing a variety of different lipid subclasses, and significant differences in these lipid molecules were detected among different groups, highlighting variations in lipid metabolism patterns. High levels of PC and TG might play a protective role by maintaining membrane integrity, while lower levels of AcCa suggest the possibility of enhancing antioxidant capacity through glycolytic pathways. Overall, this study offers novel insights into the application of curcumin in the domain of semen preservation and screens out potential biomarkers during the preservation process.
Reviewer 3 Report
Comments and Suggestions for Authors
See the attached file

Comments on the Quality of English LanguageSee the attached file.
Author Response
Response to Reviewer 3 Comments
Thank you very much for taking the time to review this manuscript. Your advice is very important to the manuscript. We have revised the manuscript following your recommendations and hope that our revised manuscript will meet your request and can be published on IJMS. Here is a response to your suggestion.
Abstract
The abstract is too crowded (496 words, instead of a maximum of 250 words), which might overwhelm the reader. The information presented here should be concise.
- The term ROS isused multiple times, leading to redundancy. It would be more concise to mention ROS once and use alternatives like “oxidative stress” or “oxidative damage” in subsequent references.
- The abstract does not mention whether theresults were statistically significant. Including this information would strengthen the credibility of the findings.
- Theresults section includes many specific numbers (for instance, 820 metabolites, 250 and 93 unique metabolites, 49 lipid subclasses, 2,495 lipid molecules), which might overwhelm the reader. Summarizing key findings orfocusing on the most critical data might clarify the results.
A:Thanks very much for your advice. We first modified the repeated occurrence of ROS so that the summary looks more concise. Second, we modified the results included in the summary and explained the statistical significance (P <0.05), which clearly emphasizes the credibility of the results. Finally, we chose some of the numbers presented in the results, showing some important results that allow the reader to get the information more quickly. We keep only the important results as much as possible in the abstract, but want to present all the results as much as possible in the abstract, which will cause the abstract to look more, so we modify the whole abstract, hoping that this can satisfy your opinion.
Results
Line 126: The word ‘time’ is redundant; consider removing it.
A:Thank you for your suggestions. I checked the manuscript, I have deleted the ‘time’.
Lines 128-131: The authors should paraphrase this sentence for clarity. Suggestion: After a 24-hour period of storage, nonotable variations were observed in the plasma membrane integrity rate among the remaining groups, except for the groupexposed to 25 µmol/L, which showed a statistically significant difference (p < 0.05).
A:Thank you for your suggestions. We have modified it following your recommendations.
Line 276: This sentence should be rewriten for clarity. Suggestion: The number of DAMs in each pathway was counted, and enrichment factors and P values were shown.
A:Thank you for your suggestions. We have modified it following your recommendations.
Discussion
This section should ideally discuss this study's results and interpret or link them to previous studies and their hypotheses. It should end with the implications of the study and highlight its limitations. The authors should also suggest future research directions.
The authors presented some sections of the discussion as if they were a literature review, and the study's results were notdiscussed in some cases. They presented too much material, becoming too verbose, making the flow of information difficult to follow. I advise the authors to reduce the information presented and present only relevant information to thisstudy.
A: Thank you for your advice. We have revised the discussion according to your suggestion to include wordy, repetitive sections. At the same time, some expressions are not clear.
Materials and Methods
The experimental design should ideally be a subheading on its own. More information should have been provided on theexperimental animals, such as their housing and nutrition, as these will greatly impact the parameters that are evaluated.
This section is too verbose and tends to overwhelm your readers. Ideally, it should be a concise description of what you did to collect the analysed data. Some sections could be rewriten, highlighting just the basic steps.
Do you think the sample size of 15 is adequate to draw conclusions from this study? Justify this number in view of itsstatistical power.
A:Thank you for your advice. We supplement the experimental design at the beginning of the MM following your suggestion and included it separately as a subtitle. At the same time, we have made a concise description of the parts to make it easier for readers to understand.
Our experimental base is the black goats used in this experiment are all rams. The goal is also to use AI to expand the population size. We also refer to the relevant literature, judged our sample size as statistically significant according to the number of samples in other studies, and obtained the corresponding results in this study. But we understand your concern that we will increase the sample size as much as possible in the follow-up trial to make the results more complete. The references are as follows:
Sun, P.; Zhang, G.; Xian, M.; Zhang, G.; Wen, F.; Hu, Z.; Hu, J. Proteomic Analysis of Frozen–Thawed Spermatozoa with Different Levels of Freezability in Dairy Goats. Int. J. Mol. Sci. 2023, 24, 15550.
Menezes, E.B., Velho, A.L.C., Santos, F. et al. Uncovering sperm metabolome to discover biomarkers for bull fertility. BMC Genomics 20, 714 (2019).
Zhang, L.; Wang, Y.; Sohail, T.; Kang, Y.; Niu, H.; Sun, X.; Ji, D.; Li, Y. Effects of Taurine on Sperm Quality during Room Temperature Storage in Hu Sheep. Animals 2021, 11, 2725.
Ji, K.; Wei, J.; Fan, Z.; Zhu, M.; Yuan, X.; Zhang, S.; Li, S.; Xu, H.; Ling, Y. Preservative Effects of Curcumin on Semen of Hu Sheep. Animals 2024, 14, 947.
Zhao, G.; Zhao, X.; Bai, J.; Dilixiati, A.; Song, Y.; Haire, A.; Zhao, S.; Aihemaiti, A.; Fu, X.; Wusiman, A. Metabolomic and Transcriptomic Changes Underlying the Effects of L-Citrulline Supplementation on Ram Semen Quality. Animals 2023,
Sun X, Zhang L, Kang Y, Wang X, Jiang C, Wang J, Sohail T, Li Y: Alpha-lipoic acid improves the quality of ram spermatozoa stored at 4°C by reducing oxidative stress and increasing mitochondrial potential. Frontiers in veterinary science 2024, 10.
Conclusions
- Theconclusion could be more concise. For example, the statement
“Phosphatidylcholine, fatty acid, and acylcarnitine subclass molecules show varying levels of expression amongdifferent groups” could be streamlined to focus on the key takeaway regarding their role in preservation.
- The conclusion could be strengthened by suggesting potential applications or implications of the findings. Forexample, how might the results inform future research or practical applications in preserving goat semen or otherbiological materials?
- Whilethe conclusion mentions that curcumin preserved the integrity of the plasma membrane and acrosome, and suppressed ROS production, it would be useful to provide a brief explanation of the mechanisms by which curcuminachieves these effects. This would enhance the depth of your conclusion.
- The statement that phosphatidylcholine, fatty acid, and acylcarnitine subclass molecules are valuable in preservation is worthy of note but somewhat unclear. Providing additional detail on how these lipids add to preservation and the
implications of their varying levels of expression would make this point more impactful.
A:Thank you for your suggestions. Our team has revised its conclusions section.
In conclusion, the supplementation of 25 µmol/L Cur during semen preservation notably enhanced the sperm quality (including sperm motility, motor performance, plasma membrane integrity, and acrosomal integrity), antioxidant capacity (increased activities of T-AOC, CAT, and SOD, etc.), and energy metabolism (levels of MMP and ATP) of Hainan Black sheep. Decreased levels of ROS and MDA were also observed. Further studies indicated that Cur could regulate metabolites such as malic acid, niacin, leucine, and isoleucine in the seminal plasma of black goats to influence metabolic processes like the citric acid cycle, cholesterol metabolism, fatty acid metabolism, and cAMP signaling pathway, thereby ensuring semen quality. Simultaneously, the lipid composition of preserved sperm was analyzed, revealing a variety of different lipid subclasses, and significant differences in these lipid molecules were detected among different groups, highlighting variations in lipid metabolism patterns. High levels of PC and TG might play a protective role by maintaining membrane integrity, while lower levels of AcCa suggest the possibility of enhancing antioxidant capacity through glycolytic pathways. Overall, this study offers novel insights into the application of curcumin in the domain of semen preservation and screens out potential biomarkers during the preservation process.
Round 2
Reviewer 3 Report
Comments and Suggestions for Authors
See the attached file.

Author Response
Thank you very much for your reviewing of the manuscript! We believe that all your suggestions are critical to improving the content of the manuscript. We revised the manuscript and indicate it in yellow highlights. It is hoped that the revised manuscript can meet the requirements for publication. Here is a peer-to-peer response to your proposal:
- Line 26: The word ‘oxitadive‘ is misspelt. Check and do the needful.
A:Thank you for your reminder! We checked and found that we had a spelling error. We have made the changes.
- Lines 190-192: There are some typo errors where some words are joined together. The authors should carefully examine and separate them.
A:Thank you for your advice! We have made the changes to your request.
- Line 384: This sentence needs to be paraphrased to make it grammatically correct. Suggestion: Semen possesses some antioxidant capacity; among these capacities, the first-line defence mechanism called the enzyme triad is the most effective.
A:Thank you for your advice! We have made the changes to your request.
- Line 389-393: The sentence appears too long, and the authors should break it into two sentences.
A:Thank you for your advice! We have made the changes to your request.
- Lines 405-406: The authors mentioned that several studies have explored the impacts of nano-formulations of diverse herbal ex-tracts (MENFs, TENFs, and CENFs) on the cryopreservation quality of goat semen. Some of these studies should be cited here.
A:We have examined this part of the manuscript. We have already cited the relevant literature in the relevant discussion (17). This discussion is derived from this literature.
- The authors didn’t adhere to the MDPI referencing format.
A:We ran a unified format for the references.